# The Role of the Pharmacist in Selecting the Best Choice of Medication Formulation in Dysphagic Patients

**DOI:** 10.3390/jpm12081307

**Published:** 2022-08-12

**Authors:** Guendalina Zuccari, Sara Macis, Silvana Alfei, Leonardo Marchitto, Eleonora Russo

**Affiliations:** 1Department of Pharmacy, University of Genoa, Viale Cembrano, 16148 Genoa, Italy; 2Hospital Pharmacy, Department Technical Health, San Paolo Hospital, Via Genova, 17100 Savona, Italy; 3Department of Sciences for the Quality of Life, University of Bologna, Corso D’Augusto 237, 47921 Rimini, Italy

**Keywords:** hospital pharmacy, dysphagic patients, off-label prescriptions, solid oral formulations, compounding pharmacist, patient safety

## Abstract

Usually, the administration of drugs by feeding tube in dysphagic patients involves handling of marketing licenses outside their term, due to the lack of suitable formulations. This circumstance has put health professionals in the dilemma of choosing the formulation whose manipulation possibly does not alter the effectiveness of the drug. In this regard, a practical guide providing indications on the prescription, handling, and administration of drugs through enteral feeding tube could be of paramount utility. For this purpose, we have considered the 1047 solid oral pharmaceutical forms included in the formulary of San Paolo Hospital (Savona, Italy). From our analysis, it emerges that 95% of medicinal products are worryingly used off-label and 40% have to be managed by the hospital pharmacists without having suitable indications by either the manufacturers or by literature studies. To fill this gap, we have compiled a detailed table containing missing indications derived from pharmacist expertise and evidence-based practices, with the aim that the sharing of our procedures will contribute to make uniform pharmacological therapies from one hospital to another. This study will allow doctors to have easy access to information on drugs that can be prescribed and nurses to become familiar only with the pharmaceutical forms that can be administered.

## 1. Introduction

In the past, interest in enteral nutrition (EN) was rather low, due to the introduction of new and safer parenteral administration techniques. However, in the last two decades, EN through percutaneous endoscopic gastrostomy (PEG) has been reevaluated as a valid metabolic rebalancing therapy, both by virtue of an improvement in the quality of the tubes and nutritional mixtures used. Additionally, thanks to the recent dietary and physiological acquisitions of the digestive system, a more correct overview of the limitations of total parenteral nutrition (TPN) has been achieved [1,2,3]. According to the guidelines of the European Society for Clinical Nutrition and Metabolism (ESPEN) on EN in adults [4], in all the conditions in which an indication for artificial nutrition (AN) is present and where there is normal functioning of the gastrointestinal tract with the possibility of covering needs by the enteral route, EN should be considered the first choice nutritional technique. In fact, it has been demonstrated that EN is simpler, more physiological less expensive, and safer than TPN, since it avoids the risks of venous catheter infections [5]. Furthermore, it is therapeutically superior to TPN, as the nutrients perform a direct trophic action on the intestinal lumen by inducing the release of trophic hormones and supporting the structural and functional integrity of the intestinal mucosa, which seems essential for maintaining immune function [6]. In our territory, managed by the local health authority ASL 2 Savonese (Liguria, Italy), the data are confirming this trend, as from 2019 to 2021, the hospitalized patients, both adults and pediatrics, requiring EN were 706, and 142 were EN patients in affiliated nursing homes (RSA) and 202 those at home from 2021 to April 2022.

In clinical practice, the feeding tube is also necessarily used as a route of administration for pharmacological therapies, because this route is less risky for systemic infections (septicemia) and more economical with respect to parenteral drug administrations [5]. Even considering that injectable formulations have the great advantage of ensuring the total absorption of the drug and do not require any formulation manipulations, they are not suitable for long-term use [2]. On the contrary, the administration of drugs by the feeding tube usually involves handling of marketing licenses outside their term (off-label use), due to the inadequacy of medicines on the market for this route of administration [6]. Consequently, to avoid undesired therapeutic failures, health care professionals must face daily the dilemma of choosing formulations whose manipulation does not alter drug efficacy. Although the use of enteral feeding tubes has become widely employed for the administration of both nutrients and pharmacological therapies, the lack of standardized guidelines and protocols at both national and international level is a limit to the homogeneity and appropriateness of administrations and prescriptions. Therefore, for patients subjected to EN, there are particular situations where the administration of oral medicines can be a pharmaceutical challenge.

Consequently, this study arose from the necessity to accomplish a proper therapy for dysphagic adult and pediatric patients, whose needs are usually unmet by the pharmaceutical companies. In this context, the role of the hospital pharmacist is essential in selecting and compounding the medicinal product able to ensure the best outcomes of the pharmacological therapy. In the case of patients subjected to EN, it becomes impossible to keep the medicinal product intact, since it is often necessarily crushed or diluted. Meanwhile, it would be mandatory to guarantee that such manipulations do not affect the pharmacokinetics of the drug, and that no interaction between the drug and the nutritional mixture could occur. The essential need to ensure a therapy for dysphagic patients too often pushes health-care professionals to seek impromptu solutions, in which it may happen that not all the necessary pharmaceutical items have been evaluated. In fact, the guidelines still refer to general behavior and general principles, without directing towards specific protocols. In this scenario, off-label drug administrations—sometimes adequate, others often incorrect—have been justified.

From these considerations, it is clear there is a need for shared procedures deriving from the fusion between clinical practice and what exists in the field of academic research. The aim of our study was to provide a practical guide on the correct use of solid oral formulations for patients subjected to EN through PEG. With this purpose, practical suggestions for the prescription, methods of handling, and administration of drugs through enteral feeding tube not only in hospitals but also at home have been provided in form of a reader-friendly and easily consultable table. We report advice received from manufacturers, collected from the literature, or deriving from the experience born within the pharmacy of the hospital. Particularly, we firstly asked pharmaceutical companies to share information regarding active pharmaceutical ingredient (API) stability following manipulation of the formulation and change of route of administration. Secondly, we analyzed the medicines managed by the local health authority ASL 2 Savonese, providing information about their suitability for administration by tube. In addition, extemporaneous liquid formulations of some drugs to be administered through the enteral tube are proposed, which can be of valuable help in order to make therapies uniform and to improve the quality of medical practice. Finally, we have also outlined the procedure for hospital staff to administer pharmacotherapy by tube.

## 2. Materials and Methods

The medicinal products considered were those available in the district area of Liguria (Italy), according to the local formulary PTRO (Prontuario Terapeutico Ospedaliero Regionale), and according to the minor formulary ASL 2 Savonese. The list of medicines analyzed comprises 1047 solid oral pharmaceutical forms, equal formulations but with a different dosage from the same company, have been grouped together in the same line, thus obtaining a list of 701 formulations. The information regarding the splittability/crushability of the considered medicinal products was obtained by evaluating the summary of product characteristics (SmPC) of the drug (source: Italian Medicines Agency AIFA, Agenzia Italiana del Farmaco database) [7] and in particular by analyzing qualitative and quantitative composition, pharmaceutical form, posology, and method of administration.

Meantime, 124 pharmaceutical companies were contacted with a specific letter, requesting detailed information on the API’s stability upon crushing of the tablets or opening of the capsules (1), on suitability of mortar for crushing (2), and on possible recommendations for API administration via tube or alternative routes (3) (Figure 1).

Other eminent sources considered were the practical guidelines from the Società Italiana di Nutrizione Artificiale e Metabolismo (SINPE) and European Society for Clinical Nutrition and Metabolism (ESPEN). In particular, the SINPE guideline on Nutrizione Artificiale Ospedaliera [8] and the ESPEN guideline on home enteral nutrition [9]. In addition, since administration via tube is a route that in most cases is not provided for in the SmPC, some information has been extrapolated by consulting the Handbook of Drug Administration via Enteral Feeding Tubes on behalf of the British Pharmaceutical Nutrition Group [10] and reading published studies present in the literature (PubMed database). Finally, indications of use were established also by analyzing the list of excipients employed by the producer. In these instances, the directions provided are based on the pharmacist’s practical experience and knowledge. The sources chosen to evaluate the suitability of drug administration via tube are summarized in Table 1 in hierarchical order.

For each solid oral drug formulation listed in the hospital formulary, the Anatomical Therapeutic Chemical (ATC) Classification System, the name of the API, the trade name, the manufacturer, the pharmaceutical form, if the product can be administered via tube or not, instructions for handling, and finally the sources of the information are highlighted in different colors according to their level of reliability. An attempt was made to provide as much information as possible: the events that the administration by tube can be carried out, when to administer the drug, compatibility between the drug and EN, and the presence of some particular excipients (e.g., alcohol).

Since the use of oral liquid pharmaceutical forms creates a minimal risk of tube obstruction and ensures adequate absorption of the drug, a switch from a solid oral formulation to a corresponding one in liquid form was considered advisable whenever possible. However, the liquid formulations may be hyperosmolar and require appropriate dilution. The volume of water (*V*) to be added has been calculated with the following formula:(1)$$V_{f}=V_{i}\times \frac{\frac{mOsm_{initial}\;}{kg}}{\frac{mOsm_{desired}\;}{kg}}$$
(2)$$V=V_{f}-V_{i}$$
where *mOsm_desired_* is 300–500 mOsm/kg, *mOsm_initial_* is the osmolality of the pharmaceutical product, *V_i_* is the volume of the liquid pharmaceutical product before handling, and *V_f_* is the volume of the final product with the desired osmolality.

To complete our study, we reported the most numerous extemporaneous liquid preparations intended to be administered by PEG to patients in the Neurology, Internal Medicine, and Otorhinolaryngology departments of the San Paolo Hospital in Savona.

## 3. Results and Discussion

### 3.1. Drug Formulations for Administration by Tube: The Pivotal Role of the Pharmacist

The therapeutic process is defined as the sum of several phases, including drug procurement, storage, preservation, stock management, prescription, preparation, administration, and the evaluation of side effects and/or benefits (Italian Ministry of Health, recommendation n. 7 for the prevention of death, coma, or severe damage arising from errors in drug therapy) [11]. Such definition evidences that there may be numerous factors that can lead to an unreliable therapy, and meanwhile health-care professionals must be able to ensure that the hospitalized patients receive the appropriate drug in the correct pharmaceutical form and route of administration. Specifically, the skills of the hospital pharmacist, in the context of preventing errors in therapy, could play a key role in evaluating the correct and safe use of drugs, both in the prescription and in the administration phase. In fact, it has been already demonstrated that pharmacy intervention can significantly reduce the number of errors related to administering medication through the enteral feeding tube with minimal additional workload [12].

Nowadays, the need to carry out a correct manipulation of oral drugs to be administered to patients subjected to EN is an increasingly emerging problem. Before deciding whether to modify the original pharmaceutical form in any way, the compounding pharmacist refers to SmPC, but, unfortunately, the clinical studies on the bioavailability of oral pharmaceutical forms performed by the companies rarely consider administration via artificial tubes. In most cases, manufacturers lack data to support the chemical stability of tablets when crushed and dispersed in water prior to use, and there are no studies comparing the efficacy or bioavailability of split and crushed solid pharmaceutical forms, which means that no company feels able to provide recommendations on the use of drugs by means of a tube. In this study, we contacted 124 pharmaceutical companies to obtain more detailed information on drug stability after crushing or switching routes of administration. Unfortunately, only one company (0.8%) gave us further information. In most cases, companies only suggested referring to the SmPC. Indeed, drug stability assessment performed by pharmaceutical companies generally involves the testing of the drug substance or drug product using a stability-indicating method in order to establish the retest period (for premarket stability) and shelf life (for commercial stability). Since the companies had not carried out any evaluation of the stability of the drug after handling and/or change of administration route, they avoided taking any kind of responsibility in giving us advice. However, before resorting to off-label use, the general indications stress first of all evaluation of the existence of formulations suitable for alternative routes to oral administration. Table 2 shows APIs for which there are multiple options of routes of administration.

When such alternatives are not suitable, as indicated in the column of limitations, or only oral formulations are available, the PEG also becomes a unique access route for drug administration. In these cases, the SINPE guideline for Hospital Artificial Nutrition states that the administration of drugs by enteral tube preferably requires the use of liquid pharmaceutical forms, when available, or as a last resort the trituration of the solid forms and their dispersion in a suitable solvent [8]. It must be considered that switching from one formulation to another may involve dose adjustment and careful monitoring of the plasma levels of the drug for imperfect equivalence between the different formulations (e.g., liquid phenytoin vs. tablets) [13]. This raises a number of issues regarding the responsibility of such administrations and of any adverse effects that the patients may experience.

Many members of health-care teams, especially pharmacists, are in a position to raise the awareness of potential drug–nutrient interactions and incompatibilities that may derive from the off-label use of a drug. Pharmaceutical incompatibility arises in dosage forms where the release of API is controlled by enteric or delayed coatings. In these cases, the manipulation (i.e., trituration) of the medicinal product leads to an immediate release and absorption of the API with overdose at the start of treatment and subtherapeutic dosages thereafter. As shown in Figure 2, modified-release tablets are the fourth most numbered formulation and all the modified-release solid oral formulations, including gastroresistant ones, represent the 12% of the total medications included in the Savona hospital formulary.

Pharmacological incompatibility manifests itself with an alteration in pH, motility, or gastrointestinal secretions resulting from a pharmacological effect of the drug and generally alters the tolerance or absorption of enteral nutrition and/or the absorption of other drugs administered in polypharmacy. It is the case of drugs with anticholinergic effect that relax smooth muscle and inhibit gastric motility, such as antihistamines (promethazine, diphenhydramine, etc.), tricyclic antidepressants (amitriptyline, imipramine, maprotiline, trimipramine, etc.), phenothiazines (chlorpridomazine, etc.), and antiparkinsonians (biperidene, trihexyphenidyl, bornaprine, orphenadrine, etc.).

Pharmacokinetic interactions can occur following the simultaneous administration of drugs and EN blends since the nutritional mixtures can alter the processes of release, absorption, distribution, metabolism, and excretion of drugs, which, in turn, can alter the kinetics of nutrients. Clinically, the most significant drug–enteral feeding interactions concern phenytoin and carbamazepine. The greatest number of compatibility studies with EN preparations have been performed for phenytoin [14]. The plasma levels of the drug were lower (70–80%) when administration was associated with EN, while they increased when EN was interrupted 2 h earlier and restored 2 h later. The mechanisms proposed were different, from the joining of the drug to the proteins or electrolytes of the nutrition mixture to the alteration of the solubility. It was observed that the interaction was more evident when phenytoin was introduced directly at the jejunal level, perhaps due to a decrease the time of intestinal transit. It is well known that the absorption of theophylline varies according to the composition of the diet; therefore, to avoid a 60–70% decrease in drug absorption, due to the increased metabolism that some diets (rich in proteins and low in carbohydrates) cause, it is necessary to suspend enteral feed at least 1 h before administration and restore it 2 h after. For the sake of completeness, we point out that some drugs are optimally absorbed only on an empty stomach (e.g., tetracyclines, penicillins, rifampicin, verapamil, atenolol, captopril).

Pharmacodynamic interactions occur when alterations in the action of the drug are related to pharmacological antagonism at the receptor site. The vitamin K content of nutritional preparations can determine an antagonism towards the therapeutic action of oral anticoagulants, such as warfarin. Initially, it was thought that this antagonism was only to be attributed to the quantity of vitamin K contained in the nutritional mixtures and therefore preparations with low doses of the same (less than 75–78 mg/1000 kcal) were recommended. Recently, this antagonism has also been observed for preparations with minimal contents of vitamin K and therefore the decrease in the therapeutic effect of the drug has been related to an alteration of its absorption probably attributable to its union with soy proteins and caseinates. This makes monitoring the prothrombin time necessary to guarantee the anticoagulant effect, as well as the indication to prefer heparin in critical clinical conditions [15].

### 3.2. The Feasibility of Drug Administration by Enteral Feeding Tube

#### 3.2.1. Liquid Formulations

The best choice of a formulation to be administered via tube could be represented by a liquid one. Indeed, the liquid formulations can be easily administered with sufficient tranquility by means of a tube and generally they are immediately diluted in gastric juices and promptly absorbed. However, the assumption that liquid formulations are the first choice may be questionable. Cosolvents, such as ethanol, glycerol, and propylene glycol, may be present in all drug formulations being used and the acceptable daily intake (ADI) may be easily exceeded. Therefore, before shifting from a solid dosage form to a liquid, it is recommended to consider this worrying issue, i.e., ranitidine syrup (Ranidil 150 mg/10 mL) [16].

Another noteworthy aspect relates to the osmolality of liquid formulations, which is one of the physical characteristics that most affect individual tolerance to a preparation. Osmolality values close to those of intestinal secretions are better tolerated (100–400 mOsm/kg H_2_O) [13]. Formulations having osmolality values higher than 6000 mOsm/kg H_2_O) if administered without dilution could cause intolerance phenomena, especially if introduced at high speed or with tubes located in the duodenum or jejunum. This type of interaction is sometimes misinterpreted as the symptoms tend to be attributed to an intolerance to nutritional support or to gastrointestinal infections resulting in an incongruous interruption of the enteral feeding. Sorbitol is a case in point. Sorbitol is a common excipient of liquid formulations, often used as a stabilizer and sweetener. High amounts (>10 g/day) can cause intraluminal production of gas and abdominal tension, and, at higher doses (>15 g/day), important secondary effects, such as abdominal spasms and diarrhea. Pharmaceutical forms with a high sorbitol content include iron protein succinylate oral solution (Rekord Ferro 40 mg/15 mL) [17] and acyclovir oral suspension (Aciclovir Dorom 400 mg/5 mL) [18]. In cases where antiviral therapy at 400 mg five times a day is required, it is easy to exceed sorbitol ADI. Mannitol may provide the same undesirable reactions. This often happens when using the oral route for medicinal products with exclusive parenteral indication. In routine practice, when the osmolality of the preparations is not known, it is advisable to dilute the medicine with at least 30 mL of water to make the administration in the stomach compatible, thus preventing diarrheic phenomena by osmotic effect. Moreover, some suspensions and syrups may be too viscous and lead to the obstruction of the tube, such as clotrimoxazole, amoxicillin–clavulanic acid, which should always be diluted with at least 100–150 mL of water and introduced through tubes of appropriate diameter. These drugs, in fact, in addition to the risk of obstruction, have a high tube-crossing time that leads to a delay in the administration time, resulting in a possible decrease in the dose effectively administered.

Also, formulations in drops are frequently hyperosmolar (e.g., clonazepam, digoxin), and the amount of water used for drug dilution must then be obviously counted in the water–electrolyte balance. In some cases, their administration is not recommended for the adhesion of the drug to the plastic walls of the tube, resulting in therapeutic ineffectiveness (e.g., diazepam [19], carbamazepine suspension [20]). Some other liquid medical solutions cause gastric motility to slow down, and the formation of insoluble gelatins in the presence of acids predispose to the formation of bezoars (indigestible deposits that can form in the stomach). In patients in coma, sedated or with altered state of consciousness and with gastroesophageal reflux, such aggregates can cause obstruction of the esophagus with sometimes impossibility of removal of the enteral probe itself (e.g., liquid sucralfate) [21].

#### 3.2.2. Solid Formulations

Recommendation n.19 by the Italian Ministry of Health deals with the risks related with the crushing and splitting of oral solid formulations [22]. Capsules and tablets require necessarily manipulation for their administration via PEG tube. In addition, particular attention should be paid to the choice of enteral mixtures, because those characterized by a high protein concentration, especially containing caseinates, interact with numerous drugs, and, due to the high viscosity, cause clots obstructing the tube. Once again, to avoid tube obstruction, it is the responsibility of the pharmacist to carefully evaluate the components of the formulation (Figure 3).

Regarding the methods of administration via PEG, it should be noted that the infusion technique is very influential. In fact, the continuous-release mode (even if conducted with pumps) more frequently predisposes to clogging of the tubes because it does not allow washing between meals, as well as not leaving free margins for drug absorption in the fasting state. Therefore, bolus and intermittent techniques are preferable. For flushing the tube, no solution has been shown to be superior to water in preventing occlusion. Obstruction may arise also from inadequately crushed tablets, precipitate formation caused by interaction between feed and drug formulation, or between drugs, and consequently it is always suggested to give each API separately, avoiding the fixed formulations [23,24]. In addition, many known interactions between drugs and conventional foods are also applicable to EN. For example, amoxicillin and digoxin are adsorbed to the fiber contained in nutritional mixtures, aluminum salts induce the precipitation of dietary proteins, tetracyclines and ciprofloxacin form complexes with calcium, paracetamol is adsorbed by pectins, and phenytoin and warfarin bind to proteins [25].

### 3.3. Analysis of the Solid Oral Formulations Present in the Hospital Formulary

From the analysis of the solid oral formulations of the 701 drugs included in the hospital formulary, 339 (48%) cannot be crushed or administered via tube; however, for 211 of these (30%) alternative dosage forms or other routes of administration are possible (Figure 4). For the remaining 52% of active ingredients, however, their suitability for administration via PEG was found, but only for 5% of these, the possibility of crushing the tablet into a fine powder or opening the capsule in patients unable to swallow is indicated in the SmPC. This means that 95% of the drugs are given outside the terms of their product license.

Medication management is not so obvious. It often represents an area in which there can be many differences from one hospital to another due to the lack of standardized protocols. At San Paolo Hospital in Savona, the key role played by the hospital pharmacist is particularly evident from Figure 5, where the pharmacist’s expertise in administering a feeding tube therapy is necessary for the management of 40% of the formulations present in the hospital formulary. These data arose from the analysis of Table 3, which contains embedded all the collected information. Particularly, the different colors, both in Figure 5 and in Table 3, are indicative of the sources behind the decisions on the feasibility of using a solid pharmaceutical form administered by tube, and each color refers to a level of reliability, as reported in Table 1.

The practical suggestions embedded in Table 3 are partly available in the SmPC or were supplied by the companies at our request (green lines), partly present in the literature (pink lines), partly suggested by guidelines (light blue lines), and partly developed in the hospital pharmacy according to pharmacists’ knowledge (yellow lines). When the SmPC does not include information about the change in administration route, the suitability of a formulation to be administered via tube was taken, firstly considering the SINPE and ESPEN guidelines, where it is clearly stated that it is dangerous to break up prolonged-release drugs (retard formulations) or gastroresistant preparations, whose manipulation can cause overdoses or reduction of the therapeutic effect. Moreover, if the drug is in form of soft capsules, it is not possible to crush them nor is it advisable to pierce them to suck the content. The guidelines also suggest switching from the oral solid dosage form to the corresponding liquid form where possible and to administer one drug at a time. If any of these conditions are present, the pharmacist has to act accordingly (light blue lines). Instead, pink lines are representative of drugs, whose administration via tube was already experienced and reported in the literature, while the yellow colored lines report all cases where the pharmacist’s experience is essential to the preparation of the medicinal product. The importance of the pharmacist’s expertise in the administration of a therapy by the feeder tube is evident. Indeed, 40% of medicinal products have to be managed by the pharmacist without any data provided by the manufacturer, research studies, or evidence-based practices present in the literature.

### 3.4. Magistral Liquid Preparations

The analysis of the prescriptions within the hospital shows that liquid pharmaceutical forms are administered only to a small extent, despite being the most suitable for this route of administration. This low percentage of liquid forms can be partly explained by the lack of a wide availability of liquid medicinal products in the hospital formulary. As for the pharmaceutical form in most cases, tablets were prescribed, followed by liquid formulations (solutions, suspensions, and drops), powders for oral solutions, and capsules. To increase the administrations of liquid forms and meanwhile reduce the use of off-label prescriptions, the pharmacists are used to compounding some magistral preparations (Table 4).

### 3.5. Instructions for the Nursing Staff Concerning Administration of Drugs via Tube

The administration of drug therapy should be a unitary act, performed by the same person. It is necessary to avoid, as far as possible, interruptions during the preparation and administration of drugs. The main steps of the procedure can be listed as follows:if the patient is undergoing EN, temporarily stop the infusion;before administration, wash the probe with 30 mL of water;take, where necessary, a tablet crusher, wash it with water and dry it, then grind the tablet to a fine powder;put the powder or other pharmaceutical forms directly in a plastic cup, add 30 mL of water at room temperature, shake and dissolve (possibly with the help of a disposable plastic spoon);once a solution or a suspension is obtained, draw it with a 60 mL catheter cone syringe (dedicated syringe for EN). Make sure there are no drug residues left in the glass;check the correct positioning of the tube; insert the syringe cone into the tube connector and flush the medication dose down the feeding tube. If the medical prescription provides for the administration of several drugs at the same time, do not simultaneously grind several drugs and do not mix them in the same syringe, but it is necessary to rinse the tube between one drug and another with at least 5–10 mL of water to ensure that the tube is clean during the transition to the next drug;rinse the tube with at least 40 mL of water after the administration;restart the feed, unless a break is required.

A thorough description of the recommendations for each pharmaceutical form is summarized in Table 5.

## 4. Conclusions

The present paper has highlighted the need to oversee and investigate the problem of administering drugs by feeding tube in dysphagic patients. The management of therapeutic pathways, due to the complexity and high clinical risk that it entails, requires advanced, updated, and integrated skills. The alteration of oral pharmaceutical forms, if not properly managed, can lead to errors in therapy, side effects, occupational exposure by inhalation or contact with the drug, and cost increase. The advice of the clinical pharmacist, especially for the verification of alternative pharmaceutical forms, can prove effective in reducing therapeutic errors. This study provides health-care teams a practical guide for the correct administration of oral drugs in patients undergoing EN through PEG. The hope that has accompanied the drafting of this document is that it can be a tool that allows the prescriber to receive the correct information on the drug and consequently to adapt the pharmacological prescription for the new type of administration. Secondly, we want to provide nurses with practical suggestions regarding the correct methods of handling and adequate administration of drugs via tube, in order to ensure continuity of care and adherence to the therapeutic program.

## Figures and Tables

**Figure 1 jpm-12-01307-f001:**
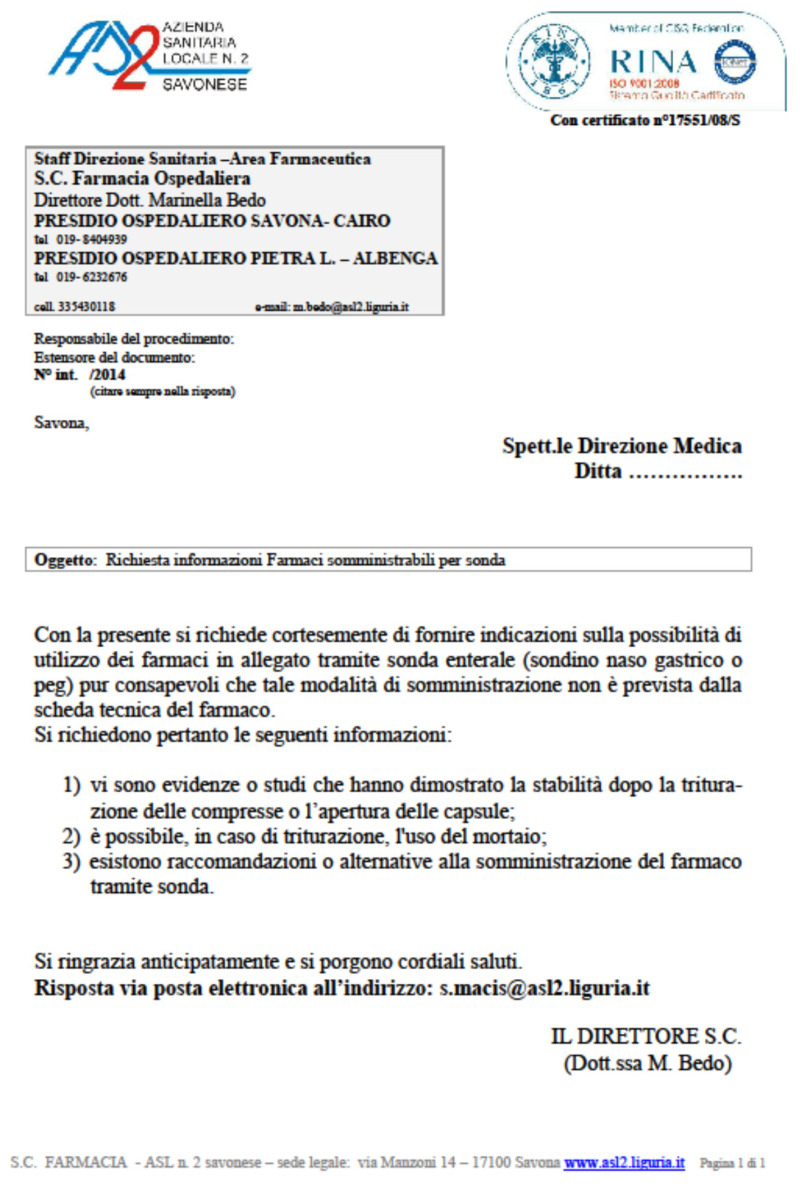
Letter addressed to the pharmaceutical companies whose products appear on the hospital formulary.

**Figure 2 jpm-12-01307-f002:**
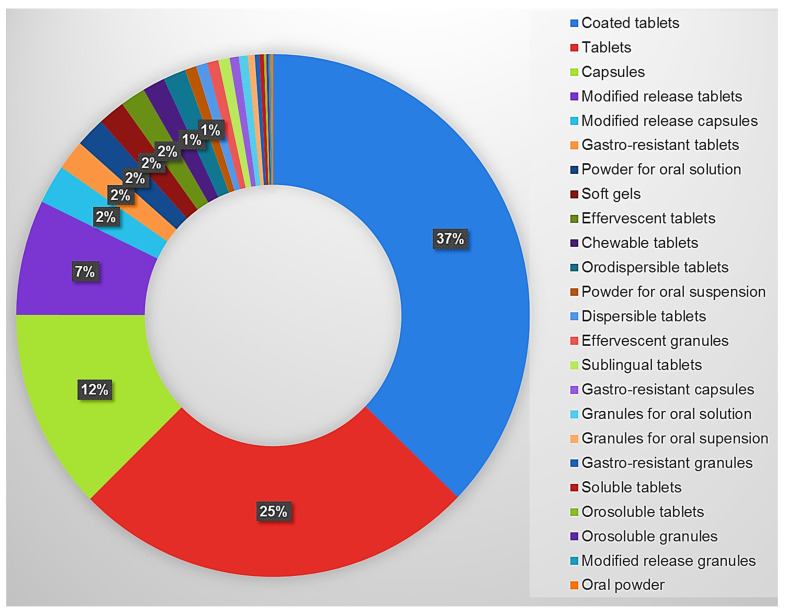
Solid oral dosage forms included in the Savona hospital formulary.

**Figure 3 jpm-12-01307-f003:**
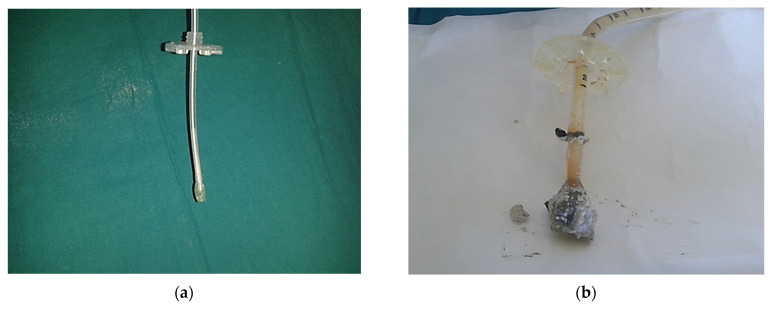
(**a**) Obstruction of a tube. (**b**) Obstruction of a PEG with breakage of the terminal bumper.

**Figure 4 jpm-12-01307-f004:**
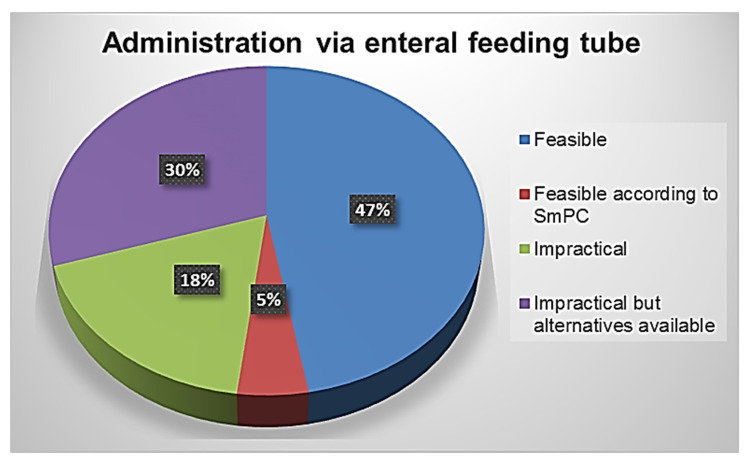
Feasibility of drug administration via enteral feeding tube of the solid oral dosage forms included in the Savona hospital formulary.

**Figure 5 jpm-12-01307-f005:**
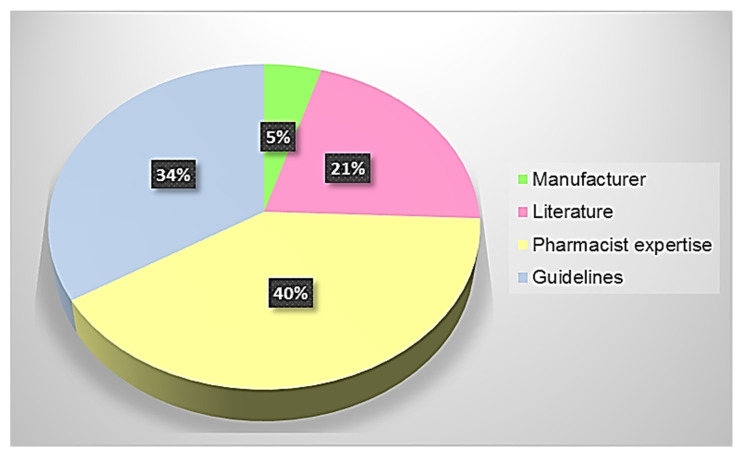
The weight of pharmacist skill in the administration of a therapy by feeder tube. The different colors are indicative of the sources behind the decisions on the feasibility of using a solid pharmaceutical form administered by tube.

**Table 1 jpm-12-01307-t001:** Sources chosen to evaluate the suitability of drug administration via tube and their level of reliability in decreasing order. The sources of the information are highlighted in different colors according to their level of reliability. Green, level A (SmPC); light blue, level B (Guidelines); pink, level C (literature); yellow, level D (pharmacist know-how).

Source	Level of Reliability
Summary of product characteristics	A
European and Italian guidelines on artificial nutrition	B
Research studies available in the literature	C
Hospital pharmacist know-how	D

**Table 2 jpm-12-01307-t002:** APIs endowed with formulations alternative to oral administration.

Administration Route	API	Limitations
Intramuscular (i.m.)	Ampicillin/sulbactam Lorazepam Prednisolone	Expensive Invasive procedure Inappropriate in immunocompromised or bleeding-prone patients Only feasible with expert help
Intravenous (i.v.)	Dexamethasone Digoxin Phenytoin Hydromorphone Methotrexate Morphine Prometazin	Expensive Invasive procedure Only feasible with expert help
Inhalation	Salbutamolo Nicotina Zanamivir	Unavailable, not to be used if there is trauma to the upper airway
Rectal	Aspirin Bisacodil Caffeine/ergotamin Lactulose Morphine Prochloroperazine Prometazin Sodium polystyrene sulfonate	Low compliance Inappropriate in patients with heart disease, immunocompromised, rectal surgery or bleeding
Subcutaneous	Fentanyl Hydromorphone Morphine	Expensive Invasive procedure May cause local tissue damage Absorption of the drug may be slower than via i.v or i.m. Inappropriate in immunocompromised patients Only feasible with expert help
Transdermal	Estradiol Fentanyl Nicotine Nitroglycerin	Inappropriate in patients with cutaneous rush, dermatitis or open lesions Risk of skin irritation, itching, contact dermatitis

**Table 3 jpm-12-01307-t003:** General suggestions for the oral solid medications listed in the Savona hospital formulary given by tube. The sources of the information are highlighted in different colors according to their level of reliability. Green, level A (SmPC); light blue, level B (Guidelines); pink, level C (literature); yellow, level D (pharmacist know-how).

ATC	API (Strength)	Brand Name (Manufacturer)	Dosage Form	Via Tube	Operative Information	Ref.
A02AD01	Aluminum hydroxide Magnesium hydroxide (800 mg)	MAALOX (SANOFI)	Chewable tablets	No	Interaction with dietary proteins Tube obstruction Use alternative medications	
A02AH	Sodium bicarbonate (500 mg)	SOD BIC NAR (NOVA ARGENTIA)	Tablets	No	Use other routes of administration	
A02BA02	Ranitidine hydrochloride (150 mg)	RANIDIL (A.MENARINI)	Effervescent tablets	Yes	The tablet should be diluted with at least 75 mL of water (to be avoided in patients on a low sodium diet or with phenylketonuria) Alternatives: liquid formulation to be diluted with 60 mL of water (contains ethyl alcohol), or sugar-free galenic preparation	
A02BA02	Ranitidine hydrochloride (150 mg)	ZANTAC (GLAXOSMITHKLINE)	Coated tablets	No	The effervescent tablets are the first choice (to be avoided in patients on a low sodium diet or with phenylketonuria) Alternatives: liquid formulation to be diluted with 60 mL of water (contains ethyl alcohol), or Ranitidine 15 mg/mL galenic preparation	[8,9]
		ZANTAC SOLUB (GLAXOSMITHKLINE)	Effervescent tablets			
		RANITIDINA MYL (MYLAN)	Coated tablets			
		RANITIDINA RATIO (RATIOPHARM)	Coated tablets			
		RANITIDINA TEV (TEVA)	Coated tablets			
A02BB01	Misoprostol (200 µg)	CYTOTEC 200 (PFIZER)	Tablets	Yes	Crush the tablet in a mortar, add 15 mL water, draw the suspension into the syringe and administer immediately Rinse the mortar with 15 mL water, draw this water into the syringe and administer Due to the poor stability of the API better an alternative therapy (Ranitidine or Lansoprazole)	[10]
A02BC01	Omeprazole (20 mg)	ANTRA (ASTRAZENECA)	Gastro-resistant capsules	Yes	Open the capsule, suspend the granules contained therein with 20 mL water and then administer after 15–20 min in a tube >8 Fr The granules must not be crushed For children the galenic preparation of omeprazole suspension 2 mg/mL is required	[26]
	Omeprazole (10 mg)	OMEPRAZOLO TEV (TEVA)				
A02BC02	Pantoprazole sodium sesquihydrate (20, 40 mg)	PANTORC (TAKEDA)	Gastro-resistant tablets	No	Change route of administration or substitute API with Omeprazole or Lansoprazole	[8,9]
A02BC03	Lansoprazole (15, 30 mg)	LANSOX (TAKEDA)	Orodispersible tablets	Yes	Disperse in a small amount of water and administer through a nasogastric tube or oral syringe	SmPC
A02BC03	Lansoprazole (15, 30 mg)	LANSOX (TAKEDA)	Capsules	Yes	Open the capsule, suspend the granules with water and administer after 20 min Prefer orodispersible tablets	[26]
A02BC04	Rabrepazole sodium (10, 20 mg)	PARIET (JANSSEN CILAG)	Gastro-resistant tablets	No	Do not grind gastro-resistant tablets Alternatives: esomeprazole, lansoprazole or omeprazole	[8,9]
A02BC05	Esomeprazole magnesium trihydrate (10 mg)	NEXIUM (ASTRAZENECA)	Gastro-resistant granules	Yes	To a dose of 10 mg add 15 mL water, mix and leave to thicken for a few minutes Do not crush the granules, withdraw the suspension with a syringe and inject through the tube of 6 Fr caliber or greater for 30 min Rinse the tube with 15 mL water	SmPC
A02BC05	Esomeprazole magnesium trihydrate (10 mg)	LUCEN (MALESCI)	Gastro-resistant granules	Yes	Mix the granules with 15 mL water, allow to thicken Pick up the suspension with a syringe and inject it into a tube of 6 Fr caliber or greater	SmPC
A02BX02	Sucralfate (2 g)	SUCRALFIN (SANOFI)	Granules for oral suspension	No	Risk of tube occlusion, bezoar formations	[10,21]
		SUCRAMAL (SCHARPER)				
A02BX13	Sodium alginate Sodium bicarbonate (250 + 133.5 mg)	GAVISCON (RECKITT BENCKISER)	Chewable tablets	No		[8,9]
A03AA05	Trimebutine maleate (150 mg)	DEBRIDAT (SIGMA TAU)	Soft gels	No	Alternatives: solution for injection or suppositories	[8,9]
A03AA06	Rociverin (10 mg)	RILATEN (LABORATORI GUIDOTTI)	Coated tablets	Yes	Tablet can be triturated and administered with water	
A03AX13	Simethicone (40 mg)	MYLICONGAS (JOHNSON & JOHNSON)	Chewable tablets	No	Alternative: drops to be diluted with 50 mL water and given immediately	[8,9]
A03BB01	Scopolamine methyl bromide (10 mg)	BUSCOPAN (BOEHRINGER INGELHEIM)	Coated tablets	Yes	Tablet can be triturated Alternative: solution for injection	
A03FA01	Metoclopramide hydrochloride monohydrate (10 mg)	PLASIL (SANOFI)	Tablets	No	Alternative: syrup which must be diluted and given away from EN	[8,9]
A03FA03	Domperidone maleate (10 mg)	DOMPERIDONE AGE (ANGENERICO)	Tablets	No	Alternatives: suppositories or oral suspension (to be diluted with water and administered at least 15 min before the start of EN)	[8,9]
A03FA03	Domperidone (10 mg)	PERIDON (ITALCHIMICI)	Coated tablets	No	Alternatives: suppositories or oral suspension (to be diluted with water and administered at least 15 min before the start of EN)	[8,9]
A04AA01	Ondansetron hydrochloride dihydrate (4 mg)	ZOFRAN (GLAXOSMITHKLINE)	Coated tablets	No	Prefer orodispersible tablets, syrup or solution for injection	[8,9]
A04AA01	Ondansetron (8 mg)	ZOFRAN (GLAXOSMITHKLINE)	Orodispersible tablets	Yes		
A04AA01	Ondansetron hydrochloride (4, 8 mg)	ONDANSETRONE TEV (TEVA)	Coated tablets	No	Alternative: Zofran syrup	[8,9]
A04AA02	Granisetron hydrochloride (2 mg)	KYTRIL (ROCHE)	Coated tablets	No	Alternative: solution for injection	[8,9]
A04AD12	Aprepitant (125, 80 mg)	EMEND (MSD)	Capsules	No	Open the capsule and administer the granules with water without crushing Preferred alternative: Ivemend solution for injection	
A05AA02	Ursodeoxycholic acid (450, 150 mg)	DEURSIL (CHEPLAPHARM ARZNEIMITTEL)	Modified release capsules	No	Alternatives: granules for oral suspension, capsules	[8,9]
A05AA02	Ursodeoxycholic acid (450, 150 mg)	URSILON (I.B.I.GIOVANNI LORENZINI)	Capsules	Yes	Open the capsule and suspend the granules in 20 mL water then administer immediately	[10]
A05BA03	Sylimarin (70 mg)	LEGALON (MEDA PHARMA)	Coated tablets	Yes		
A06AB06	Senna leaves (12 mg)	PURSENNID (GLAXOSMITHKLINE)	Coated tablets	No	Prefer other laxatives in syrups	[10]
A06AB58	Sodium picosulfate Magnesium oxide Anhydrous citric acid	PICOPREP (FERRING)	Powder for oral solution	Yes	Dissolve the powder in 150 mL of water	
A06AD11	Lactulose (10 g)	LAEVOLAC EPS (ROCHE)	Granules for oral solution	Yes	Dilute with at least 60 mL water Avoid high doses, risk of diarrhea and malabsorption of dietary nutrients	
A06AD15	Macrogol 4000 (4, 9.7, 10 g)	PAXABEL (IPSEN)	Powder for oral solution	Yes	Dissolve the powder in 125 mL water and administer immediately Seek dietary advice in case of chronic constipation	
		REGOLINT (LABORATORI BALDACCI)				
		LAXIPEG (ZAMBON)				
A06AD65	Macrogol 4000 Sodium sulfate Sodium bicarbonate Sodium chloride Potassium chloride (17.4, 34.8 g)	ISOCOLAN (GIULIANI)	Powder for oral solution	Yes	Dissolve the powder in 250 mL water then shake and administer immediately	SmPC
A06AD65	Macrogol 3350 Sodium bicarbonate Sodium chloride Potassium chloride (13.8 g)	MOVICOL (NORGINE)	Powder for oral solution	Yes	Dissolve the powder in 125 mL water, shake and administer immediately Seek dietary advice in case of chronic constipation	
A06AD65	Macrogol 3350 Sodium sulfate Sodium bicarbonate Sodium chloride Potassium chloride Ascorbic Acid Sodium Ascorbate (112 g)	MOVIPREP (NORGINE)				
A07AA06	Paromomycin sulfate (250 mg)	HUMATIN (PFIZER)	Capsules	No	Alternative: syrup to be diluted in at least 60 mL water and then given immediately	[8,9]
A07AA09	Vancomycin hydrochloride (250 mg)	MAXIVANIL (GENETIC)	Capsules	No	Alternatives: powder for oral solution or for infusion	[10]
A07AA11	Rifaximin (200 mg)	NORMIX (ALFA WASSERMANN)	Coated tablets	No	Alternative: oral suspension	[8,9]
A07AA12	Fidaxomicin (200 mg)	DIFICLIR (ASTELLAS)	Coated tablets	Yes	Grind the tablet, suspend the powder in water and administer immediately	[27]
A07BC05	Diosmectite (3 g)	DIOSMECTAL (MALESCI)	Powder for oral solution	Yes	Dissolve the powder in half a glass of water and then administer	
A07DA03	Loperamide hydrochloride (2 mg)	LOPERAMIDE HEX (SANDOZ)	Capsules	No	Use other dosage form	
A07DA03	Loperamide hydrochloride (2 mg)	DISSENTEN (SOC.PRO.ANTIBIOTICI)	Tablets	Yes		
A07EA06	Budesonide (3 mg)	ENTOCIR (ASTRAZENECA)	Modified release capsules	No	Alternative: suppositories	[8,9]
A07EA07	Beclomethasone dipropionate (5 mg)	CLIPPER (CHIESI FARMACEUTIC)I	Gastro-resistant tablets	No	Alternative: rectal preparations for distal ulcerative colitis	[8,9]
A07EC01	Sulfasalazine (500 mg)	SALAZOPYRIN EN (PFIZER)	Gastro-resistant tablets	No		[8,9]
A07EC02	Mesalazine (400, 500, 800, 1200 mg)	PENTASA (FERRING)	Modified release tablets	Yes	Break the tablet and let it disintegrate in water. Alternatives: suppositories or rectal suspension	SmPC
A07EC02	Mesalazine (400, 500, 800, 1200 mg)	MESAVANCOL (GIULIANI)	Modified release tablets	No	Do not crush Alternatives: rectal suspension or suppositories	[8,9]
		PENTACOL (SOFAR)				
A07FA02	Saccharomyces boulardii (5 bn)	CODEX (ZAMBON)	Capsules	No		
		CODEX (ZAMBON)	Powder for oral suspension			
A09AA02	Pancrelipase (150, 300 mg)	CREON 25000 (BGP PRODUCTS)	Modified release capsules	Yes	Open the capsule without grinding the granules, suspend them in 20 mL water or in liquids, with a pH <5.5 such as apple, pineapple juice or yogurt, then administer immediately	
A10BA02	Metformin hydrochloride (500, 850, 1000 mg)	GLUCOPHAGE (BRUNO FARMACEUTICI)	Coated tablets	Yes	Grind the tablet, suspend the powder in 20 mL water, and administer immediately	[10]
A10BA02	Metformin hydrochloride (500, 850, 1000 mg)	ZUGLIMET (ZENTIVA)	Coated tablets	Yes	Grind the tablet, suspend the powder in 20 mL water, then administer immediately	
A10BB01	Glibenclamide (5 mg)	DAONIL (SANOFI)	Tablets	Yes	Grind the tablet, suspend the powder in 10 mL water, then administer immediately Stop EN at least 30 min before drug administration	
A10BB09	Gliclazide (30, 60 mg)	DIAMICRON (SERVIER)	Modified release tablets	No		[8,9]
A10BB09	Gliclazide (80 mg)	GLICLAZIDE (ZENTIVA)	Tablets	Yes	Grind the tablet and suspend the powder in 10 mL water, then administer immediately Stop EN at least 30 min before drug administration Divisible tablet.	[10]
A10BB12	Glimepiride (2 mg)	GLIMEPIRIDE ACC (ACCORD HEALTHCARE)	Tablets	Yes	Disperse the tablet in 10 mL water until a fine dispersion is created Administer before the main meal	[10]
	Glimepiride (3 mg)	GLIMEPIRIDE SAN (SANDOZ)				
	Glimepiride(1, 2, 3, 4 mg)	AMARYL(SANOFI)				
A10BD02	Metformin hydrochloride Glibenclamide (500 + 5 mg)	GLICONORM (ABIOGEN PHARMA)	Coated tablets	No	Administer the two drugs separately Flush with 10 mL water between each one	[8,9]
	Metformin hydrochloride Glibenclamide (400 + 2.5/500 + 5 mg)	GLIBOMET (LABORATORI GUIDOTTI)				
	Metformin hydrochloride Glibenclamide (400 + 2.5 mg)	SUGUAN (SANOFI)				
A10BD05	Pioglitazone Metformin hydrochloride (15 + 850 mg)	COMPETACT (TAKEDA)	Coated tablets	No		[8,9]
A10BD07	Sitagliptin phosphate monohydrate Metformin hydrochloride (50 + 850 mg)	JANUMET (MSD)	Coated tablets	Yes		
A10BD08	Vildagliptin Metformin (50 + 1000 mg)	EUCREAS (NOVARTIS)	Tablets	No		[8,9]
A10BD10	Saxagliptin hydrochloride Metformin hydrochloride (2.5 + 850 mg)	KOMBOGLYZE (ASTRAZENECA)	Coated tablets	No		[8,9]
A10BF01	Acarbose (50, 100 mg)	GLUCOBAY (BAYER)	Tablets	Yes	The tablets do not disperse easily in water but require gentle stirring for 5 min. A fine suspension is obtained which must be administered immediately, then continue with EN	[10]
A10BG03	Pioglitazone hydrochloride (15, 30 mg)	PIOGLITAZONE ACV (ACTAVIS)	Tablets	Yes		
		ACTOS (TAKEDA)				
A10BH01	Sitagliptin phosphate monohydrate (100 mg)	TESAVEL (ADDENDA)	Coated tablets	Yes		
A10BH01	Sitagliptin Phosphate Monohydrate (50, 100 mg)	JANUVIA (MSD)	Coated tablets	Yes	Check the glycemia	
A10BH02	Vildagliptin (50 mg)	GALVUS (NOVARTIS)	Tablets	No		
A10BH03	Saxagliptin hydrochloride (5 mg)	ONGLYZA (ASTRAZENECA)	Coated tablets	No		
A10BX02	Repaglinide (0.5, 1 mg)	REPAGLINIDE EG (EG)	Tablets	Yes		
	Repaglinide (1, 2 mg)	REPAGLINIDE SAN (SANDOZ)				
A11AA03	Vitamins Mineral salts	SUPRADYN (BAYER)	Effervescent tablets	Yes	Dissolve in water and administer at the end of the effervescence	
A11CC04	Calcitriol (0.25 µg)	DIFIX (PROMEDICA)	Soft gels	No	Drug adheres to the walls of the tube, risk of loss	[8,9]
		ROCALTROL (ROCHE)				
		CALCITRIOLO TEV (TEVA)				
A11DA01	Thiamine hydrochloride (300 mg)	BENERVA (TEOFARMA)	Gastro-resistant tablets	No		[8,9]
A11DB	Thiamine hydrochloride Cyanocobalamin Pyridoxine hydrochloride	BENEXOL B12 (BAYER)	Coated tablets	No	Alternative: solution for injection	[8,9]
A11DB	Thiamine hydrochloride Cyanocobalamin Pyridoxine hydrochloride	TRINEVRINA B6 (LABORATORI GUIDOTTI)	Coated tablets	No	Alternative: solution for injection	[8,9]
A11HA02	Pyridoxine hydrochloride (300 mg)	BENADON (BAYER)	Gastro-resistant tablets	No	Alternative: solution for injection	[8,9]
A11HA03	Alpha-Tocopherol (400 IU)	RIGENTEX (BRACCO)	Soft gels	No		[8,9]
A11JB	Sodium Citrate Potassium Citrate Thiamine diphosphate Riblofavin-5 monophosphate monosodium Pyridoxine hydrochloride Citric acid	BIOCHETASI (SIGMS TAU)	Effervescent granules	Yes		
A12AA04	Calcium Carbonate (1000 mg)	CALCIODIE (SPA)	Effervescent tablets	Yes	Dissolve in 20 mL water and administer at the end of the effervescence at least 1 h before or 2 h after the NE	
A12AX	Calcium Carbonate Cholecalciferol	IDEOS (MEDA PHARMA)	Chewable tablets	Yes		
		METOCAL VIT.D3 (MEDA PHARMA)				
A12BA01	Potassium Chloride (600 mg)	KCL-RETARD (ASTELLAS)	Modified release tablets	No	Change route of administration	[8,9]
A12BA30	Potassium Citrate Potassium Succinate Potassium Malate Potassium Tartrate Potassium Bicarbonate	POTASSION (ACARPIA)	Effervescent granules	Yes	Dissolve in water, and administer at the end of the effervescence	
A16AA02	Ademetionine Butanedisulfonate (400 mg)	SAMYR 400 (BGP PRODUCTS)	Gastro-resistant tablets	No	Alternative: solution for injection	[8,9]
B01AA03	Warfarin sodium (5 mg)	COUMADIN (BRISTOL-MYERS SQUIBB)	Tablets	Yes	Grind the tablet and suspend the powder in 10 mL water, then administer immediately Stop feeding at least 1 h before and 2 h after drug administration Avoid supplementation with vitamin K Warning: drug-EN interaction, continuous monitoring and dosage adjustment. If possible administer low molecular weight heparins	[10]
B01AA07	Acenocumarol (1, 4 mg)	SINTROM (MERUS LABS LUXCO II SARL)	Tablets	Yes	Grind the tablet, dissolve in 10 mL water and administer immediately Trituration can alter bioavailability Check for clotting and prothrombin time	[10]
B01AB11	Sulodexide (250 LSU)	VESSEL (ALFA WASSERMANN)	Soft gels	No	Oily excipients do not guarantee correct administration Alternative: solution for injection	[8,9]
B01AC04	Clopidogrel besylate (75 mg)	CLOPIDOGREL AURO (ACTAVIS)	Coated tablets	No	Alternative: Plavix	
B01AC04	Clopidogrel besylate (75, 300 mg)	PLAVIX (SANOFI)	Coated tablets	Yes	Crush the tablet and disperse the powder in 10 mL water Administer by an 8 Fr tube	[10]
B01AC05	Ticlopidine hydrochloride (250 mg)	TIKLID (SANOFI)	Coated tablets	Yes	Grind the tablet, dissolve the powder in 20 mL water, and administer immediately preferably concurrently with EN to avoid gastrointestinal adverse effects	
B01AC06	Acetylsalicylic Acid (100 mg)	ACIDO ACETILSALICILICO SAN (SANDOZ)	Gastro-resistant tablets	No	The tablet must not be crushed to avoid irritating effects on the intestine Alternatives: effervescent tablets, chewable tablets, granules for oral solution	[8,9]
B01AC06	Acetylsalicylic Acid Magnesium Hydroxide Algedrate	ASCRIPTIN (SANOFI)	Tablets	Yes	Grind the tablet, dissolve the powder in 10 mL water, and administer immediately concurrently with EN	
B01AC06	Lysine Acetylsalicylate (75, 100, 160 mg)	CARDIRENE (SANOFI)	Powder for oral solution	Yes	Dilute with at least 60 mL of water and administer immediately in conjunction with EN	
B01AC07	Dipyridamole (75, 200 mg)	PERSANTIN (BOEHRINGER INGELHEIM)	Coated tablets	Yes		[10]
B01AC07	Dipyridamole (75 mg)	PERSANTIN R (BOEHRINGER INGELHEIM)	Modified release capsules	No		[8,9]
B01AC10	Indobufen (200 mg)	IBUSTRIN (PFIZER)	Tablets	Yes		
B01AC22	Prasugrel hydrochloride (100 mg)	EFIENT (DAIICHI SANKYO)	Coated tablets	Yes	Grind the tablet and administer immediately Highly photosensitive hygroscopic drug	
B01AC24	Ticagrerol (90 mg)	BRILIQUE (ASTRAZENECA)	Coated tablets	Yes	Crush the tablet to a fine powder, transfer it in half a glass of water and administer Rinse the tube with water	SmPC
B01AC30	Dipyridamole Acetylsalicylic Acid (200 + 25 mg)	AGGRENOX (BOEHRINGER INGELHEIM)	Modified release capsules	No	The granules can block the tube Alternatives: change dosage form or route of administration	
B01AE07	Dabigatran etexilate mesylate (75, 110, 150 mg)	PRADAXA (BOEHRINGER INGELHEIM)	Capsules	No	The opening of the capsule increases drug bioavailability by 75% Bleeding risk Switch to another anticoagulant	
B01AF01	Rivaroxaban (10, 15, 10 mg)	XARELTO (BAYER)	Coated tablets	Yes	Crush the tablet, dissolve the powder in water, then administer immediately Rinse the tube with water The drug administration should be followed immediately by EN	SmPC
B01AF02	Apixaban (2.5, 5 mg)	ELIQUIS (BRISTOL-MYERS SQUIBB)	Coated tablets	Yes	Crush the tablet and suspend the powder in 60 mL water or 5% dextrose water solution, then administer immediately The crushed tablets are stable up to 4 h when stored at 30 °C	SmPC
B02BX05	Eltrombopag olamine (25, 50 mg)	REVOLADE (NOVARTIS)	Coated tablets	No		
B03AA01	Ferrous (II) glycine sulphate (100 mg)	NIFEREX (UCB PHARMA)	Capsules	No	Risk of tube occlusion	
B03AA03	Ferrous (II) gluconate (80 mg)	PRONTOFERRO (IBSA FARMACEUTICI)	Effervescent tablets	Yes	Dissolve the tablet in water, administer before EN Alternatives: ampoules or syrup	SmPC
B03AA07	Ferrous (II) sulfate (105 mg)	FERROGRAD (TEOFARMA)	Modified release tablets	No	Alternative: ampoules	[8,9]
B03BB01	Folic Acid (5 mg)	FOLINA (TEOFARMA)	Capsules	No	Alternative: liquid formulation	[8,9]
B06AA	Promelase (30 mg)	FLAMINASE (GRUNENTHAL)	Gastro-resistant tablets	No		[8,9]
C01AA05	Digoxin (0.0625, 0.125 mg)	LANOXIN (ASPEN PHARMA)	Tablets	Yes	Grind the tablet, dissolve it in 10 mL of water, administer immediately. Stop NE 2 h before and 2 h after the drug administration. Inaccurate dosage, change in bioavailability: monitor the patient Preferably give liquid formulation	[10]
C01AA05	Digoxin (0.1, 0.2 mg)	EUDIGOX (TEOFARMA)	Soft gels	No	Alternative: liquid formulation	[8,9]
C01AA08	Methyl Digoxin (0.1 mg)	LANITOP (RIEMSER PHARMA)	Tablets	Yes	Grind the tablet, dissolve it in 10 mL water and administer immediately Preferably give drops	
C01BA	Dihydroquinidine hydrochloride (250 mg)	IDROCHINIDINA R (TEOFARMA)	Capsules	Yes	Open the capsule, suspend the powder in 10 mL water, then administer immediately	
C01BC03	Propafenone hydrochloride (325 mg)	RYTMONORM (BGP PRODUCTS)	Modified release capsules	No	Alternatives: Rytmonorm coated tablets or solution for injection	[8,9]
C01BC03	Propafenone hydrochloride (150, 300 mg)	RYTMONORM (BGP PRODUCTS)	Coated tablets	Yes	Grind the tablet and dissolve the powder in 10 mL water, then administer immediately Interaction with EN: the effect can be enhanced by the presence of food Alternative: solution for injection	
C01BC04	Flecanide Acetate (100 mg)	ALMARYTM (MEDA PHARMA)	Tablets	Yes	Disperse the tablet in 10 mL water for 2 min The suspension must be administered immediately	[10]
C01BC04	Flecainide Acetate (100 mg)	FLECAINIDE TEV (TEVA)	Tablets	Yes	Divisible tablets	
C01BD01	Amiodarone hydrochloride (200 mg)	CORDARONE (SANOFI)	Tablets	Yes	Divisible tablets	
C01BD07	Dronedarone hydrochloride (400 mg)	MULTAQ (SANOFI)	Coated tablets	Yes		
C01DA08	Isosorbide dinitrate (5 mg)	CARVASIN (TEOFARMA)	Sublingual tablets	No	Do not grind Reduction of the absorption of the drug due to the hepatic first pass effect Administer sublingually only if the patient is conscious	[8,9]
C01DA14	Isosorbide mononitrate (60 mg)	DURONITRIN (ASTRAZENECA)	Modified release tablets	No	Alternative: Nitroglycerine transdermal formulation	[8,9]
	Isosorbide mononitrate (60 mg)	MONOKET MULTITAB (CHIESI FARMACEUTICI)	Modified release tablets			
	Isosorbide mononitrate (20 mg)	MONOKET (CHIESI FARMACEUTICI)	Modified release tablets			
	Isosorbide mononitrate (50 mg)	MONOKET RETARD (CHIESI FARMACEUTICI)	Modified release capsules			
	Isosorbide mononitrate (20, 40, 50, 80 mg)	MONOCINQUE R (IST.LUSOFARMACOI)	Modified release capsules			
C01EB17	Ivabradine hydrochloride	PROCORALAN (SERVIER)	Coated tablets	Yes	Crush the tablet, suspend in water and administer immediately	
	Ivabradine hydrochloride	CORLENTOR (STRODER)				
C01EB18	Ranolazine (375, 500, 750 mg)	RANEXA (A.MENARINI)	Modified release tablets	No		[8,9]
C02AB01	Methyldopa (250 mg)	ALDOMET (IROKO PRODUCTS)	Coated tablets	No	Change antihypertensive drug	
C02AC01	Clonidine hydrochloride	CATAPRESAN (BOEHRINGER INGELHEIM)	Tablets	Yes	Grind the tablet, dissolve the powder in 10 mL water and administer immediately	[10]
C02CA04	Doxazosin mesylate (2, 4 mg)	CARDURA (PFIZER)	Tablets	Yes	Grind the tablet, dissolve the powder with 10 mL sterile water and administer immediately Monitor the patient and adjust the therapy	[10]
C02DC01	Loniten (5 mg)	MINOXIDIL (PFIZER)	Tablets	Yes	The tablet is dispersed in water to give a fine suspension which must be administered immediately	[10]
C02KX01	Bosentan monohydrate (62.5, 125 mg)	TRACLEER (ACTELION PHARMA)	Coated tablets	Yes	Crush the tablet and dissolve it in at least 50 mL water, it is not very soluble Use individual protections for handling Dispersible tablets (Tracleer 32 mg) for pediatrics	
C02KX02	Ambriesentan (5 mg)	VOLIBRIS (GLAXOSMITHKLINE)	Coated tablets	No		
C02KX04	Macitentan (10 mg)	OPSUMIT (ACTELION PHARMA)	Coated tablets	Yes	Crush the tablet into a fine powder and add water, to have a suspension (drug not soluble) Rinse the tube several times after administration Use individual protections for handling Teratogen drug	
C02KX05	Riociguat (2.5 mg)	ADEMPAS (MSD)	Coated tablets	Yes	The tablet can be crushed and mixed with water	SmPC
C03AA03	Hydrochlorothiazide (25 mg)	ESIDREX (NOVARTIS)	Tablets	Yes		
C03BA04	Chlorthalidone (25 mg)	IGROTON (AMDIPHARM)	Tablets	Yes	The tablet disperses in water within 2 min	[10]
C03BA08	Metolazone (5, 10 mg)	ZAROXOLYN (TEOFARMA)	Tablets	Yes	Grind the tablet, dissolve it with 10 mL water and administer immediately Monitor the patient, drug bioavailability variable	[10]
C03CA01	Furosemide (25 mg)	FUROSEMIDE (LAB.FARMACOLOGICO MILANESE)	Tablets	Yes	Grind the tablet, dissolve it in 10 mL water and administer immediately preferably concomitantly with EN to minimize gastrointestinal effects It is recommended to use the liquid form	
	Furosemide (25, 500 mg)	LASIX (SANOFI)				
	Furosemide (500 mg)	FUROSEMIDE TEV (TEVA)				
C03CA04	Torasemide (10 mg)	TORASEMIDE TEV (TEVA)	Tablets	Yes	Divisible tablet	
C03DA01	Spironolactone (100 mg)	ALDACTONE (SANOFI)	Coated tablets	Yes	Grind, dissolve the powder in 10 mL water and administer immediately Pay attention to concomitant administration with potassium.	[10]
	Spironolactone (25 mg)	ALDACTONE (SANOFI)	Capsules	Yes	Open the capsule, dissolve in 10 mL of water and administer immediately Pay attention to concomitant administration with potassium.	
C03DA01	Spironolactone (25 mg)	URACTONE SPA (SOC.PRO.ANTIBIOTICI)	Tablets	Yes		
C03DA03	Canrenone (50, 100 mg)	LUVION (THERABEL GIENNE PHARMA)	Tablets	No	Alternative: solution for injection	
C03EA01	Amiloride hydrochloride Hydrochlorothiazide (5 + 50 mg)	MODURETIC (MSD)	Tablets	Yes	Grind, dissolve in 10 mL of water and administer immediately.	[10]
C03EA14	Potassium Canrenoate Butizide (5 + 50 mg)	KADIUR (THERABEL GIENNE PHARMA)	Tablets	No		[8,9]
C03EB01	Furosemide Spironolactone (25 + 37 mg)	LASITONE (SANOFI)	Capsules	No	Administer the two drugs separately Flush with 10 mL water between each one	[8,9]
C04AD03	Pentoxifylline (400 mg)	TRENTAL (SANOFI)	Modified release tablets	No		[8,9]
C05CA53	Diosmin Hesperidin (500 mg)	DAFLON (SERVIER)	Coated tablets	Yes	Grind the tablet, dissolve the powder in 10 mL of water and administer it immediately	
C05CX	Escin (40 mg)	EDEVEXIN (I.B.I.GIOVANNI LORENZINI)	Coated tablets	No	Alternatives: solution for injection or cutaneous gel	
C07AA05	Propranolol hydrochloride (40 mg)	INDERAL (ASTRAZENECA)	Tablets	Yes	Grind the tablet, dissolve the powder in 10 mL water and administer immediately Alternative: galenic preparation (Propranolol 2 or 5 mg/mL oral suspension)	
C07AA07	Sotalol hydrochloride (80 mg)	RYTMOBETA (BGP PRODUCTS)	Tablets	Yes	Grind the tablet, dissolve the powder in 10 mL water by shaking for 5 min, then administer immediately Stop EN 2 h before and 2 h after drug administration Follow the same administration schedule	[10]
C07AA12	Nadolol (80 mg)	NADOLOLO SFV (SANOFI)	Tablets	Yes	Grind the tablet, dissolve the powder in 10 mL water and administer immediately If possible, use another beta blocker available in liquid form	
C07AB02	Metoprolol tartrate (100 mg)	LOPRESOR (DAIICHI SANKYO)	Coated tablets	Yes	It is recommended to use a beta blocker liquid preparation If Metoprolol is non-replaceable, prepare an extemporaneous suspension of Metoprolol 10 mg/mL with simple syrup, or grind the tablet and dissolve it in 10 mL of water.	[10]
		METOPROLOLO HEX (SANDOZ)	Tablets			
C07AB03	Atenolol (100 mg)	ATENOLOLO RAT (RATIOPHARM)	Coated tablets	Yes	Grind the tablet, dissolve the powder in 10 mL water under shaking for 5 min, then administer immediately Stop EN 30 min before and 30 min after drug administration Divisible tablets	[10]
		ATENOLOLO HEX (SANDOZ)			Grind the tablet, dissolve the powder in 10 mL water, then administer immediately Stop EN 30 min before and 30 min after drug administration	[10]
C07AB07	Bisoprolol fumarate (10 mg)	CONCOR (BRACCO)	Tablets	Yes	Grind the tablet, dissolve the powder in 10 mL water under shaking for 5 min, then administer immediately preferably in the morning after EN	[10]
	Bisoprolol hemifumarate (2.5 mg)	CONGESCOR (DAIICHI SANKYO)	Coated tablets	Yes	Grind the tablet, dissolve the powder in 10 mL water under shaking for 5 min, then administer immediately preferably in the morning after EN	[10]
	Bisoprolol hemifumarate (1.25, 2.5, 3.75, 5, 10 mg)	CARDICOR (RECORDATI)	Coated tablets	Yes	Grind the tablet, dissolve the powder in 10 mL water under shaking for 5 min, then administer immediately preferably in the morning after EN Divisible tablet	[10]
C07AB12	Nebivolol hydrochloride (5 mg)	NEBIVOLOLO SAN (SANDOZ)	Tablets	No	Alternative: Atenolol	
C07AG01	Labetalol hydrochloride (100 mg)	TRANDATE (TEOFARMA)	Tablets	No		
C07AG02	Carvedilol (6.25 mg)	OMERIA (MEDIOLANUM FARMACEUTICI)	Tablets	Yes	Grind and dissolve the tablet in 10 mL water Administer with EN to reduce the risk of orthostatic hypotension	[10]
C07AG02	Carvedilol (25 mg)	Carvedilol ZTV (ZENTIVA)	Tablets	Yes	Grind and dissolve the tablet in 10 mL water Administer with EN to reduce the risk of orthostatic hypotension	[10]
C08CA01	Amlodipine besylate (5, 10 mg)	NORVASC (PFIZER)	Tablets	Yes	Dissolve the tablet in 10 mL water and administer immediately concurrently with EN to minimize the gastrointestinal side effects	[10]
	Amlodipine maleate (10 mg)	AMLODIPINA WIN (ZENTIVA)				
C08CA02	Felodipine (5 mg)	FELODIPINA WPI (ZENTIVA)	Modified release tablets	No		[8,9]
C08CA05	Nifedipine (20,30,60 mg)	ADALAT CRONO (BAYER)	Modified release tablets	No	Do not grind, use the liquid formulation (Nifedicor 20 mg/mL drops) to be diluted with 60 mL water and administer immediately	[8,9]
	Nifedipine (10 mg)	ADALAT (BAYER)	Soft gels	No	Alternative: Nifedicor 20 mg/mL drop solution to be diluted with 60 mL of water	
	Nifedipine (30 mg)	CORAL (SO.SE.PHARM)	Modified release tablets	No	Alternative: Nifedicor 20 mg/mL drop solution to be diluted with 60 mL of water	
C08CA06	Nimodipine (30 mg)	NIMOTOP (BAYER)	Coated tablets	Yes	Photosensitive drug, use amber syringe Not compatible with PVC tube	
C08CA09	Lacipidine (4 mg)	LACIPIL (GLAXOSMITHKLINE)	Coated tablets	No	Drug poorly soluble in water and photosensitive Divisible tablets Alternative: Amlodipine formulations	
C08CA11	Manidipine hydrochloride (20 mg)	IPERTEN (CHIESI FARMACEUTICI)	Tablets	No		
		MANIDIPINA MYL (MYLAN)		Yes		
		MANIDIPINA TEV (TEVA)		Yes	Divisible tablets	
C08CA13	Lecardipine hydrochloride (10 mg)	ZANEDIP (RECORDATI)	Coated tablets	Yes	Administer away from EN Alternative: Amlodipine formulations	
C08DA01	Verapamil hydrochloride (40, 80, 180 mg)	ISOPTIN (BGP PRODUCTS)	Coated tablets	Yes		[10]
C08DA01	Verapamil hydrochloride (120 mg)	ISOPTIN R (BGP PRODUCTS)	Modified release tablets	No	Do not grind Alternative: solution for injection	[8,9]
		VERAPAMIL HEX (SANDOZ)				
C08DB01	Diltiazem hydrochloride (120 mg)	ALTIAZEM (IST.LUSOFARMACO)	Modified release tablets	No	Alternative: 60 mg Diltiazem divisible tablets dosage form	[8,9]
	Diltiazem hydrochloride (300 mg)	ALTIAZEM (IST.LUSOFARMACO)	Modified release capsules			
	Diltiazem hydrochloride (200, 300 mg)	TILDIEM (SANOFI)	Modified release tablets			
	Diltiazem Hydrochloride (60, 120 mg)	TILDIEM (SANOFI)	Modified release tablets			
C09AA01	Captopril (25, 50 mg)	CAPTOPRIL RAT (RATIOPHARM)	Tablets	Yes	Grind, dissolve the tablet in 10 mL water and administer immediately Stop EN at least two h before drug administration. Divisible tablets	[10]
C09AA02	Enalapril maleate (5, 20 mg)	ENAPREN (MSD)	Tablets	Yes	Grind the tablet, dissolve the powder in 10 mL sterile water and administer immediately	[10]
C09AA03	Lisinopril dehydrate (5, 20 mg)	ZESTRIL (ASTRAZENECA)	Tablets	Yes	Divisible tablets It is possible to prepare an extemporaneous galenic solution/suspension: stable for 30 days at 2–8 °C	[10]
C09AA03	Lisinopril dehydrate (20 mg)	LISINOPRIL WPI (ZENTIVA)	Tablets	Yes		
C09AA04	Perindopril arginine (5 mg)	COVERSYL (SERVIER)	Coated tablets	Yes		
C09AA04	Perindopril tosilate (5 mg)	PERINDOPRIL TEV (TEVA)	Coated tablets	No	Switch to another ACE inhibitor	
C09AA05	Ramipril (10 mg)	QUARK (POLIFARMA)	Tablets	Yes	Dissolve the tablet in 20 mL water and administer immediately Divisible tablets	[10]
	Ramipril (2.5, 5, 10 mg)	TRIATEC (SANOFI)				
C09AA13	Moexipril hydrochloride (15 mg)	FEMIPRES (UCB PHARMA)	Coated tablets	No		[10]
C09AA15	Zofenopril calcium (30 mg)	BIFRIL (IST.LUSOFARMACO)	Coated tablets	Yes		
C09BA04	Perindopril arginine Indapamide (2.5 + 0.625/5 + 1.25 mg)	PRETERAX (SERVIER)	Coated tablets	Yes	Grind, dissolve the tablet in 10 mL water and administer immediately	
	Perindopril arginine Indapamide (5 + 1.25 mg)	PRELECTAL (STRODER)				
C09BA05	Ramipril Piretanide (5 + 6 mg)	PRILACE (SANOFI)	Coated tablets	No	Divisible tablets	[8,9]
C09BA06	Quinapril hydrochloride hydrochlorothiazide (20 + 12.5 mg)	ACEQUIDE (RECORDATI)	Coated tablets	Yes	Crushable and divisible tablets	
C09BA09	Fosinopril sodium Hydrochlorothiazide (20 + 12.5 mg)	FOSICOMBI (A.MENARINI)	Tablets	No	Switch to another ACE inhibitor	[8,9]
C09CA01	Losartan Potassium (50 mg)	LORTAAN (MSD)	Coated tablets	Yes	Grind, dissolve the tablet in 10 mL water and administer immediately	[10]
C09CA03	Valsatrtan (40 mg)	TAREG (NOVARTIS)	Coated tablets	Yes		
	Valsatrtan (80, 160 mg)	NOVARTIS	Capsules			
C09CA04	Ibersartan (150, 300 mg)	APROVEL (SANOFI)	Tablets	Yes	The tablet is dispersed in 10 mL water under stirring for 5–10 min No blockage risk with an 8 Fr tube	
	Ibersartan (150 mg)	KARVEA (SANOFI)			Stirring to disperse the tablet	
C09CA06	Candesartan Cilexetil (8 mg)	RATACAND (ASTRAZENECA)	Tablets	Yes	Grind the tablet, dissolve the powder in 10 mL water and administer immediately	[10]
	Candesartan Cilexetil (16 mg)	BLOPRESS (TAKEDA)			Grind the tablet, dissolve the powder in 10 mL water and administer immediately. Alternative: Ibersartan	
	Candesartan Cilexetil (8, 16 mg)	CANDESARTAN TEV (TEVA)			Grind the tablet, dissolve the powder in 10 mL water and administer immediately Divisible tablets.	
C09CA07	Telmisartan (20, 80 mg)	PRITOR (BAYER)	Tablets	Yes	Dissolve the tablet in 5 mL water under stirring and administer immediately Hygroscopic drug	[10]
	Telmisartan (20, 40, 80 mg)	MICARDIS (BOEHRINGER INGELHEIM)				
C09CA08	Olmesartan medoxomil (10, 20 mg)	OLPRESS (MENARINI)	Coated tablets	No	Alternative: Ibersartan	
	Olmesartan medoxomil (10, 20, 40 mg)	PLAUNAC (MENARINI)				
C09DA08	Olmesartan medoxomil Hydrochlorothiazide (20 + 12.5 mg)	PLAUNAZIDE (MENARINI)	Coated tablets	No	Administer the two drugs separately Flush with 10 mL water between each one	[8,9]
	Olmesartan medoxomil Hydrochlorothiazide (20 + 25/40 + 25 mg)	OLPREZIDE (MENARINI)				
C09DB02	Olmesartan medoxomil Amlodipine besylate (40 + 5 mg)	SEVIKAR (IST.LUSOFARMACO)	Coated tablets	No	Alternative: administer separately another angiotensin II antagonist drug (eg Irbesartan) and the calcium channel blocker	[8,9]
C09XA02	Aliskiren hemifumarate (150, 300 mg)	RASILEZ (NOVARTIS)	Coated tablets	Yes		
C10AA01	Simvastatin (10, 20 mg)	SINVACOR (MSD)	Coated tablets	Yes	Grind the tablet, dissolve the powder in 10 mL water and administer immediately, preferably together with the last course of EN	[10]
	Simvastatin (10, 20 mg)	SIMVASTAT.RAT (RATIOPHARM)			Grind the tablet, dissolve the powder in 10 mL water and administer immediately, preferably together with the last course of EN Divisible tablets	
	Simvastatin (10 mg)	SIMVASTAT.TEV (TEVA)			Grind the tablet, dissolve the powder in 10 mL water and administer immediately, preferably together with the last course of EN Divisible tablets	
	Simvastatin (40 mg)	SIMVASTAT.ZTV (ZENTIVA)			Grind the tablet, dissolve the powder in 10 mL water and administer immediately, preferably together with the last course of EN.	
C10AA03	Pravastatin sodium (20 mg)	PRAVASTAT.RAT (TEVA)	Tablets	Yes	Divisible tablets	[10]
C10AA04	Fluvastatin sodium (80 mg)	LESCOL (NOVARTIS)	Modified release tablets	No	Alternative: Fluvastatin 20 or 40 mg capsules, or Atorvastatin or Pravastatin	[8,9]
C10AA05	Atorvastatin calcium trihydrate (10, 20, 40 mg)	TORVAST (PFIZER)	Coated tablets	Yes	The tablet is dispersed within 2–5 min in 10 mL water Photosensitive drug Absorption not affected by the presence of food	
C10AA07	Rosuvastatin calcium (5, 10, 20 mg)	CRESTOR (ASTRAZENECA)	Coated tablets	Yes	Disperse the tablet in water, after 5 min a milky pink dispersion is obtained, then administer immediately Better to use Atorvastatin	[10]
C10AB04	Gemfibrozil (600, 900 mg)	LOPID (PFIZER)	Coated tablets	Yes	Crushable tablets	
C10AB05	Fenofibrate (145 mg)	FULCROSUPRA (BGP PRODUCTS)	Coated tablets	No		
	Fenofibrate (200 mg)	FENOFIBRATO SAN (SANDOZ)	Capsules	No		
C10AC01	Cholestyramine hydrochloride (4 g)	QUESTRAN (BRISTOL-MYERS SQUIBB)	Powder for oral suspension	No	Risk of occlusion of the tube due to the formation of a semi-solid mass Possible interference with the absorption of other drugs	
C10AX06	Polyenoic Omega (ethyl esters of polyunsaturated fatty acids) (1000 mg)	SEACOR (SOC.PRO.ANTIBIOTICI)	Soft gels	No		[8,9]
C10AX09	Ezetimibe (10 mg)	EZETROL (MERCK SHARP & DOHME)	Tablets	Yes	Disperse the tablet in 10 mL water under stirring Administer the suspension immediately	
C10BA02	Ezetimibe Simvastatin (10 + 10 mg)	INEGY (MERCK SHARP & DOHME)	Tablets	No	Administer the two drugs separately Flush with 10 mL water between each one	[8,9]
	Ezetimibe Simvastatin (10 + 20 mg)	VYTORIN (NEOPHARMED GENTILI)				
D01BA02	Terbinafine hydrochloride (250 mg)	TERBINAFINA SAN (SANDOZ)	Tablets	Yes	Disperse the tablet under stirring in 10 mL water for 5 min, to give a fine dispersion that must be immediately administered Divisible tablets Be careful when handling	[10]
D05BB02	Acitretin (10, 25 mg)	NEOTIGASON (AUROBINDO)	Capsules	Yes	Open the capsule, disperse the content in water and administer immediately together with EN If possible change the drug	
D10BA01	Isotretinoin (10 mg)	ISOTRETINOINA DIF (DIFA COOPER)	Soft gels	No		[8,9]
D11AH04	Alitretinoin (30 mg)	TOCTINO GLAXOSMITHKLINE	Soft gels	No		[8,9]
G02AB01	Methylergometrine maleate (0.125 mg)	METHERGIN (NOVARTIS)	Coated tablets	No	Alternative: solution for injection	
G02CB01	Bromocriptine mesylate (2.5 mg)	PARLODEL (MEDA PHARMA)	Tablets	Yes	Administer together with EN	[10]
G02CB03	Cabergoline (0.5 mg)	DOSTINEX (PFIZER)	Tablets	No		
G03AA10	Ethinylestradiol Gestodene	GINODEN (BAYER)	Coated tablets	No	Alternative: transdermal patches	[8,9]
		ESTINETTE (EFFIK)				
G03AD01	Levonorgestrel (1.5 mg)	NORLEVO (LABORATOIRE HRA PHARMA)	Tablets	No	Alternative: transdermal patches	[8,9]
G03DA04	Prgesterone (100, 200 mg)	PROGEFFIK (EFFIK)	Capslues	No	Alternatives: insert the capsule vaginally or use the solution for injection	[8,9]
G03DB04	Nomegestrol acetate (5 mg)	LUTENYL (RATIOPHARM)	Tablets	No		
G03DC02	Norethisterone acetate (10 mg)	PRIMOLUT NOR (BAYER)	Tablets	Yes	Disperse the tablet in 10 mL water under stirring for 5 min, to give a fine dispersion that must be immediately administered Be careful when handling	[10]
G03HA01	Cyproterone acetate (50 mg)	ANDROCUR (BAYER)	Tablets	Yes	Grind, disperse the tablet in 10 mL water and administer immediately. Where possible switch to intramuscular administration	
G03XB01	Mifepristone (200 mg)	MIFEGYNE (EXELGYN)	Tablets	Yes		
G03XB02	Ulipristal acetate (5 mg)	ESMYA (GEDEON RICHTER)	Tablets	Yes	Crushable tablets	
G04BD	Flavoxate Propyphenazone (200 + 30 mg)	CISTALGAN (MEDA PHARMA)	Coated tablets	No		[8,9]
G04BD04	Oxybutynin hydrochloride (5 mg)	DITROPAN (SANOFI)	Tablets	Yes		
G04BD08	Solifenacin succinate (5 mg)	VESIKER (ASTELLAS)	Coated tablets	No		
G04BD12	Mirabregon (50 mg)	BETMIGA (ASTELLAS)	Modified release tablets	No		[8,9]
G04BE03	Sildenafil citrate (20 mg)	REVATIO (PFIZER)	Coated tablets	No	Alternative: oral suspension	
	Sildenafil citrate (50, 100 mg)	VIAGRA (PFIZER)	Tablets	Yes	Grind, disperse the tablet in 10 mL water and administer immediately	
G04BE08	Tadalafil (5, 10, 20 mg)	CIALIS (ELI LILLY)	Coated tablets	No		
	Tadalafil (20 mg)	ADCIRCA (ELI LILLY)				
G04CA01	Alfuzosin hydrochloride (2.5 mg)	MITTOVAL (SANOFI)	Modified release tablets	No	Alternative: Doxasozin	[8,9]
		XATRAL (SANOFI)				
G04CA01	Alfuzosin hydrochloride (2.5 mg)	XATRAL (SANOFI	Coated tablets	Yes	Alternative: Doxasozin	
G04CA02	Tamsulisin hydrochloride (0.4 mg)	PRADIF (BOEHRINGER INGELHEIM)	Capsules	No	Risk of blockage of the tube Alternative: Doxasozin	
G04CA03	Terazosin hydrochloride (5 mg)	TERAPROST (MALESCI)	Tablets	Yes		
	Terazosin hydrochloride (2, 5 mg)	TERAZOSINA TEV (TEVA)	Tablets	No		
G04CA04	Silodosin (4, 8 mg)	UROREC (RECORDATI)	Capsules	Yes	Open the capsule and suspend the content in water	
G04CB01	Finasteride (5 mg)	FINASTERIDE (NEOPHARMED GENTILI)	Coated tablets	Yes	Crushable tablets Teratogen drug	[10]
G04CB02	Dutasteride (0.5 mg)	AVODART (GLAXOSMITHKLINE)	Capsules	No	Content of the capsule liquid, irritant and teratogen	
H01BA02	Desmopressin acetate (60, 120 µg)	MINIRIN/DDAVP (FERRING)	Sublingual tablets	No	Do not crush the tablet Reduction of the absorption of the drug due to the hepatic first pass effect	[8,9]
H02AB01	Betamethasone sodium phosphate	BENTELAN (SIGMATAU)	Effervescent tablets	Yes	Dissolve the tablet in 10 mL water and administer immediately	SmPC
H02AB04	Methylprednisolone (4, 16 mg)	MEDROL (PFIZER)	Tablets	Yes	Grind the tablet, dissolve it in 10 mL water and administer immediately	[10]
H02AB07	Prednisone (5, 25 mg)	DELTACORTENE DELTACROTENE FTE (BRUNO FARMACEUTICI)	Tablets	Yes	Grind the tablet, dissolve it the tablet in 10 mL water and administer immediately	[10]
H02AB07	Prednisone (5 mg)	LODOTRA (MUNDIPHARMA)	Modified release tablets	No	Alternative: Deltacortene 5 mg	[8,9]
H02AB09	Hydrocortisone (5, 20 mg)	PLENADREN (SHIRE)	Modified release tablets	No		[8,9]
H02AB10	Cortisone acetate (25 mg)	CORTONE ACETATO (TEOFARMA)	Tablets	Yes		
H03AA01	Levothyroxine sodium (50, 100 µg)	TIROSINT (IBSA FARMACEUTICI)	Tablets	Yes	Grind the tablet, dissolve it in 10 mL water. Interaction with EN: suspend it 1 h before and resume it 2 h later drug administration Alternative: Tirosint oral solution which is not affected by simultaneous food intake	SmPC
H03AA01	Levothyroxine sodium (25, 50, 75, 100 µg)	EUTIROX (MERCK SERONO)	Tablets	Yes	Grind the tablet and dissolve it in 10 mL water, then administer immediately Suspend EN 1 h before and resume it 2 h later drug administration Monitor TSH levels Divisible tablets	[10]
	Levothyroxine sodium (25 µg)	LEVOTIROXINA TEV (TEVA)				
H03BB02	Thiamazole (Methimazole) (5 mg)	TAPAZOLE (TEOFARMA)	Tablets	Yes	Grind the tablet and disperse it in 10 mL water, then administer immediately Divisible tablets	
H05BX01	Cinacalcet hydrochloride (30, 60, 90 mg)	MIMPARA (AMGEN)	Coated tablets	No		
H05BX02	Paricalcitol (1 µg, 2 µg)	ZEMPLAR (ABBVIE)	Capsules	No	Alternative: Zemplar solution for injection	
J01AA02	Doxycycline hyclate (100 mg)	BASSADO (PFIZER)	Tablets	Yes	Grind the tablet and dissolve it in 10 mL water. Administer immediately with plenty of water to prevent irritation	[10]
J01AA08	Minocycline hydrochloride (100 mg)	MINOCIN (TEOFARMA)	Capsules	No		
J01CA01	Ampicillin (500 mg)	AMPLITAL (PFIZER)	Capsules	No	Alternative: powder for solution for injection	
J01CA04	Amoxicillin trihydrate (1 g)	ZIMOX (PFIZER)	Tablets	No	Alternatives: switch to other formulations (drops, oral suspension, powder for oral suspension)	
J01CA04	Amoxicillin trihydrate (1 g)	ZIMOX (PFIZER)	Chewable tablets	Yes	Dissolve the tablets in a half glass of water	SmPC
J01CR02	Amoxicillin trihydrate Clavulanate potassium (875 + 125 g)	ABBA (FIDIA FARMACEUTICI)	Powder for oral suspension	Yes	Dissolve the powder in 50 mL water Administer together with EN to minimize the gastrointestinal side effects	SmPC
		AUGMENTIN (GLAXOSMITHKLINE)				
J01CR02	Amoxicillin trihydrate Clavulanate potassium (875 + 125 g)	NEODUPLAMOX (VALEAS)	Coated tablets	No	Alternatives: switch to other formulations (suspension, powder for oral suspension)	[8,9]
J01CR04	Sultamicillin tosylate (750 mg)	UNASYN (PFIZER)	Coated tablets	No	Alternatives: powder for oral solution, or solution for injection	[8,9]
J01DB01	Cephalexin monohydrate (1 g)	KEFORAL (CRINOS)	Tablets	No	Switch to another cephalosporin or use the oral suspension available on the market	[8,9]
	Cephalexin (1 g)	CEPOREX (TEOFARMA)	Coated tablets			
J01DC02	Cefuroxime acetoxyethyl (250 mg)	ZINNAT (GLAXOSMITHKLINE)	Coated tablets	No	Use the syrup which must be diluted with 60 mL water and administered immediately	[8,9]
J01DD08	Cefixime (400 mg)	UNIXIME (F.I.R.M.A.)	Coated tablets	No	Alternatives: dispersible tablets or oral suspension. Stop the EN at least 1 h before and resume it 2 h after drug administration	[8,9]
J01DD13	Cefpodoxime proxetil (100, 200 mg)	CEFODOX (SCHARPER)	Coated tablets	Yes	Better to switch to the oral suspension	[8,9]
J01EE01	Trimethoprim Sulfamethoxazole (160 + 800 mg)	BACTRIM (ROCHE)	Tablets	No	Alternative: oral suspension	[8,9]
J01FA09	Clarithromycin (250, 500 mg)	KLACID (BGP PRODUCTS)	Coated tablets	No	Alternatives: oral suspension, solution for injection Photosensitive drug	[8,9]
	Clarithromycin (500 mg)	CLARITROMICINA TEV (TEVA)			Alternative: oral suspension which must be diluted with 60 mL water and administered immediately	
J01FA10	Azithromycin monohydrate (500 mg)	AZITROMICINA EG (EG)	Coated tablets	No	Alternative: oral suspension which must be diluted with 60 mL water and administered immediately It does not interact with EN	[8,9]
	Azithromycin monohydrate (500 mg)	AZITROMICINA MYL (MYLAN)			Alternative: oral suspension which must be diluted Administer 1 h before or 2 h after EN	
	Azithromycin dihydrate (500 mg)	ZITROMAX (PFIZER)			Alternative: oral suspension which must be diluted Administer 1 h before or 2 h after EN	
	Azithromycin dihydrate (500 mg)	AZITROMICINA PRG (PROGE FARM)			Alternative: oral suspension which must be diluted Administer 1 h before or 2 h after EN	
	Azithromycin monohydrate (500 mg)	AZITROMICINA SAN (SANDOZ)			Alternative: oral suspension which must be diluted Administer 1 h before or 2 h after EN	
J01FA10	Azithromycin dihydrate (500 mg)	TROZAMIL (SO.SE.PHARM)	Coated tablets	Yes	Alternative: oral suspension which must be diluted Administer 1 h before or 2 h after EN	
J01FF01	Clindamycin hydrochloride (150 mg)	DALACIN-C (PFIZER)	Capsules	Yes	Disperse the content in water and then administer Alternatives: solution for injection or other route of administration (cutaneous gel)	
J01MA02	Ciprofloxacin hydrochloride (250, 500, 750, 1000 mg)	CIPROXIN (BAYER)	Coated tablets	No	EN-drug interaction: decreased absorption and therapeutic failure Alternatives: solution for infusion or granules for oral suspension (EN must be suspended 30 min before and 30 min after drug administration)	
	Ciprofloxacin hydrochloride monohydrate (250 mg)	CIPROFLOXAC.RAT (RATIOPHARM)				
	Ciprofloxacin hydrochloride monohydrate (750 mg)	CIPROFLOXAC.SAN (SANDOZ)				
J01MA12	Levofloxacin hemihydrate (500 mg)	LEVOFLOXACINA SAN (SANDOZ)	Coated tablets	No	Alternatives: solution for injection or switch to more bioavailable Ofloxacin	
J01MA14	Moxifloxacin hydrochloride (400 mg)	AVALOX (BAYER)	Coated tablets	Yes	The coating exerts only taste masking	
J01MA17	Prulifloxacin (600 mg)	UNIDROX (ANGELINI)	Coated tablets	yes	Administer immediately after dissolution of the tablet in water	
J01XE01	Nitrofurantoin macrocrystals (50 mg)	NEOFURADANTIN (GRUNENTHAL)	Capsules	No	Risk of tube blockage	
J01XX01	Fosfomycin trometamol salt (3 g)	BERNY (SO.SE.PHARM)	Granules for oral solution	Yes	Administer away from the EN, usually at bedtime	
J01XX08	Linezolid (600 mg)	ZYVOXID (PFIZER)	Coated tablets	Yes	Grind the tablet, dissolve the powder in 20 mL water then administer immediately Prefer oral suspension or solution for infusion Any interaction with EN	[10]
J02AC01	Fluconazole (100 g)	FLUCONAZOLO EG (EG)	Capsules	Yes	Open the capsule, disperse the content in 20 mL water, then administer immediately Prefer oral suspension or solution for injection If dietary product is rich in fiber, suspend EN 1 h before and 1 h after drug administration	[10]
		DIFLUCAN (PFIZER)				
		FLUCONAZOLO SAN (SANDOZ)				
		FLUCONAZOLO HEX (SANDOZ)				
		FLUCONAZOLO RAT (TEVA)				
J02AC02	Itraconazole (100 g)	SPORANOX (JANSSEN CILAG)	Capsules	No	Alternatives: solution for infusion or oral solution to be diluted with water The absorption occurs at acidic pH Administer together with EN	
J02AC03	Voriconazole (200 mg)	VFEND (PFIZER)	Coated tablets	Yes	Grind the tablet, disperse the powder in water and administer away from EN (1 h before or 2 h after EN) Alternatives: oral suspension or powder for solution for injection	[10]
J04AB02	Rifampicin (300, 400 mg)	RIFADIN (SANOFI)	Capsules	No	Alternative: syrup to be diluted and administered away from EN	[8,9]
J04AC01	Isoniazid (200 mg)	NICOZID (PIAM FARMACEUTICI)	Tablets	Yes	Grind the tablet, dissolve the powder in 20 mL water then administer immediately Stop EN 1 h before and 2 h after drug administration	[10]
J04AK01	Pyrazinamide (500 mg)	PIRALDINA (BRACCO)	Tablets	Yes	Grind the tablet, dissolve the powder in 10 mL water then administer immediately	[10]
J04AK02	Ethambutol hydrochloride (400 mg)	ETAPIAM (PIAM FARMACEUTICI)	Coated tablets	Yes	Grind the tablet, dissolve the powder in 10 mL water then administer immediately 2 h after EN	[10]
J04AM02	Rifampicin Isoniazid (300 + 150 mg)	RIFINAH 300 (SANOFI)	Coated tablets	No	Administer the two drugs separately Flush with 10 mL water between each one	[8,9]
J04AM05	Isoniazid Pyrazinamide Rifampicin (50 + 300 +120 mg)	RIFATER (SANOFI)	Coated tablets	Yes	Suspend EN 1 h before drug administration	
J05AB01	Acyclovir (400 mg)	ACICLOVIR EG (EG)	Tablets	Yes	Grind the tablet, dissolve it in 30 mL water then administer immediately Alternatives: oral suspension or solution for injection	[10]
	Acyclovir (800 mg)	ACICLIN (FIDIA FARMACEUTICI)				
	Acyclovir (400 mg)	ACICLOVIR DRM (TEVA)				
J05AB04	Ribavirin (200 mg)	REBETOL (MSD)	Capsules	No	Alternative: Ribavirin (Rebetol) oral solution	[8,9]
		COPEGUS (ROCHE)	Coated tablets			
		RIBAVIRINA SAN (SANDOZ)	Capsules			
		RIBAVIRINA TEV (TEVA)	Capsules			
J05AB11	Valacyclovir hydrochloride (1 g)	ZELITREX (GLAXOSMITHKLINE)	Coated tablets	Yes	Better to switch to Acyclovir The tablets are difficult to crush and the powder does not suspend well Alternative: galenic suspension of Valaciclovir 50 mg/mL, by triturating the tablets and using simple syrup (good stability up to 21 days at 2–8 °C)	
		VALACICLOVIR MYL (MYLAN)				
J05AB14	Valgancyclovir hydrochloride (450 mg)	DARILIN (ROCHE)	Coated tablets	No	Alternative: Valgancyclovir oral suspension (Novir)	[8,9]
J05AE03	Ritonavir (100 mg)	NORVIR (ABBVIE)	Coated tablets	No	Alternative: Ritonavir oral suspension Tablets not crushable	[8,9]
J05AE07	Fosamprenavir calcium (700 mg)	TELZIR (VIIV HEALTHCARE)	Coated tablets	No	Alternative: Fosamprenavir oral suspension (Telzir)	[8,9]
J05AE08	Atazanavir sulfate (200, 300 mg)	REYATAZ (BRISTOL-MYERS SQUIBB)	Capsules	Yes	Open the capsule, dissolve the content in water, administer it via tube, then continue with EN	
J05AE10	Darunavir ethanolate (600, 800 mg)	PREZISTA (JANSSEN CILAG)	Coated tablets	No	Alternative: Darunavir ethanolate oral suspension	
J05AE11	Telaprevir (375 mg)	INCIVO (JANSSEN CILAG)	Coated tablets	No		
J05AE12	Boceprevir (200 mg)	VICTRELIS (MSD)	Capsules	No		
J05AE14	Simeprevir (150 mg)	OLYSIO (JANSSEN CILAG)	Capsules	No		
J05AF02	Didanosine (250 mg)	VIDEX 250 (BRISTOL-MYERS SQUIBB)	Capsules	No		
J05AF05	Lamivudine (100 mg)	ZEFFIX (GLAXOSMITHKLINE)	Coated tablets	Yes	Disperse the tablet in 10 mL water, a pale orange solution is obtained, then administer immediately Prefer: Lamivudine oral solution	[10]
J05AF05	Lamivudine (100 mg)	LAMIVUDINA MYL (MYLAN)	Coated tablets	No	Prefer liquid formulation	[8,9]
J05AF07	Tenofovir Disoproxil Fumarate (245 mg)	VIREAD (GILEAD SCIENCES)	Coated tablets	Yes	Dissolve the tablet in 100 mL water or orange juice and administer	SmPC
J05AF08	Adefovir dipivoxil (10 mg)	HESPERA (GILEAD SCIENCES)	Coated tablets	No	Switch to Tenofovir	
J05AF09	Emtricitabine (200 mg)	EMTRIVA (GILEAD SCIENCES)	Capsules	No	Alternative: Emtricitabine oral solution Increase the dose by 20% with respect to the tablets, due to reduced bioavailability of the oral solution	
J05AF10	Entecavir (0.5, 1 mg)	BARACLUDE (BRISTOL-MYERS SQUIBB)	Coated tablets	No	Switch to oral solution	[8,9]
J05AG01	Nevirapine (400 mg)	VIRAMUNE (BOEHRINGER INGELHEIM)	Tablets	No	Switch to oral suspension	[8,9]
	Nevirapine (200 mg)	NEVIRAPINA TEV (TEVA)			Alternative: liquid formulation	
J05AG03	Efavirenz (200 mg)	SUSTIVA (BRISTOL-MYERS SQUIBB)	Capsules	Yes		
J05AG03	Efavirenz (600 mg)	EFAVIRENZ MYL (MYLAN)	Coated tablets	No	Alternative: liquid formulation	[8,9]
J05AG04	Etravirine (200 mg)	INTELENCE (JANSSEN CILAG)	Tablets	Yes	Dissolve the tablet in 10 mL water and administer	SmPC
J05AG05	Rilpivirine hydrochloride (25 mg)	EDURANT (JANSSEN CILAG)	Coated tablets	No		
J05AH02	Oseltamivir phosphate (30, 45 mg)	TAMIFLU (ROCHE)	Capsules	No	Alternative: oral suspension	[8,9]
J05AR01	Lamivudine Zidovidine (150 + 300 mg)	LAMIV+ZIDOV MYL (MYLAN)	Coated tablets	No	Administer the two drugs separately in liquid form	[8,9]
J05AR02	Abacavir sulfate Lamivudine (600 + 300 mg)	KIVEXA (VIIV HEALTHCARE)	Coated tablets	No	Separate preparations of the two drugs are available in the form of oral suspension	[8,9]
J05AR03	Emtricitabine Tenofovir disoproxil (200 + 245 mg)	TRUVADA (GILEAD SCIENCES)	Coated tablets	Yes	The tablets can be dissolved in about 100 mL of water, orange juice or grape juice and taken immediately	SmPC
J05AR04	Abacavir sulfate Lamivudine Zidovudine (300 + 150 + 300 mg)	TRIZIVIR (VIIV HEALTHCARE)	Coated tablets	No	Separate preparations of the three drugs are available in the form of oral suspension	[8,9]
J05AR06	Efavirenz Emtricitabine Tenofovir disoproxil (600 +200 +245 mg)	ATRIPLA (GILEAD SCIENCES)	Coated tablets	No		[8,9]
J05AR08	Emtricitabine Rilpivirine hydrochloride Tenofovir disoproxil fumarate (200 + 25 + 245 mg)	EVIPLERA (GILEAD SCIENCES)	Coated tablets	No		[8,9]
J05AR09	Emtricitabine Tenofovir disoproxil fumarate Elvitegravir Cobicistat (200 + 300 + 150 + 150 mg)	STRIBILD (GILEAD SCIENCES)	Coated tablets	No		[8,9]
J05AR10	Lopinavir Ritonavir (200 + 50 mg)	KALETRA (ABBVIE)	Coated tablets	No	Tablet crushing reduces AUC by approximately 40% Preferred alternative: Kaletra oral solution (slightly viscous, contains alcohol and propylene glycol)	[8,9]
J05AX08	Raltegravir potassium (400 mg)	ISENTRESS (MSD)	Coated tablets	No	Drug insoluble in water	
J05AX09	Maraviroc (150, 300 mg)	CELSENTRI (VIIV HEALTHCARE)	Coated tablets	No		
J05AX12	Dolutegravir sodium (50 mg)	TIVICAY (VIIV HEALTHCARE)	Coated tablets	Yes	Grind the tablet, disperse it in water and administer away from EN (2 h before or 6 h after EN)	
J05AX14	Daclatasvir dihydrochloride (60 mg)	DAKLINZA (BRISTOL-MYERS SQUIBB)	Coated tablets	Yes	The coating serves only to cover the drug bitter taste	
J05AX15	Sofosbuvir (400 mg)	SOVALDI (GILEAD SCIENCES)	Coated tablets	Yes	The coating serves only to cover the drug bitter taste The tablets can be triturated and administered via tube concurrently with EN	
J05AX16	Dasabuvir (250 mg)	EXVIERA (ABBVIE)	Coated tablets	No	Not crushable tablet	
J05AX65	Ledipasvir Sofosbuvir (90 + 400 mg)	HARVONI (GILEAD SCIENCES)	Coated tablets	No		[8,9]
J05AX67	Ombitasvir Paritaprevir Ritonavir (12.5 + 75 + 50 mg)	VIEKIRAX (ABBVIE)	Coated tablets	No	Not crushable tablet	[8,9]
L01AA01	Cyclophosphamide (50 mg)	ENDOXAN (BAXTER)	Coated tablets	No	Alternative: solution for injection	[8,9]
L01AA02	Chlorambucil (2 mg)	LEUKERAN (GLAXOSMITHKLINE)	Coated tablets	No		
L01AA03	Melfalan (25 mg)	ALKERAN (ASPEN PHARMA)	Coated tablets	No	Alternative: solution for injection	[8,9]
L01AB01	Busulfan (2 mg)	MYLERAN (ASPEN PHARMA)	Coated tablets	No	Alternative: galenic suspension of Busulfan 2 mg/mL by triturating the tablet Shelf life 30 days if stored between 2–8 °C	
L01AX03	Temozolomide (100 mg)	TEMODAL (MSD)	Capsules	No		
	Temozolomide (100 mg)	TEMOZOLOMIDE SUH (RANBAXY ITALIA)				
	Temozolomide (5, 20, 140, 180, 250 mg)	TEMOZOLOMIDE SUH (SUN PHARMACEUTICALS)				
L01BA01	Methotrexate (2.5 mg)	METHOTREXATE (PFIZER)	Tablets	No	Alternative: Methotrexate solution for injection Tablets are divisible but not crushable	[8,9]
L01BB05	Fludarabine phosphate (10 mg)	FLUDARA (GENZYME)	Coated tablets	No	Alternative: solution for injection	[8,9]
L01BC06	Capecitabine (150, 500 mg)	CAPECITABINA ACC (ACCORD HEALTHCARE)	Coated tablets	Yes	Disperse the tablet in 200 mL warm water and then administer immediately Attention to handling, use individual protection measures	
		CAPECITABINA MYL (MYLAN)				
		XELODA (ROCHE)				
L01CA04	Vinorelbine ditartrate(20, 30 mg)	NAVELBINE (PIERRE FABRE PHARMA)	Soft gels	No	Alternative: concentrate for solution for infusion	[8,9]
L01XE01	Imatinib mesylate (100 mg)	GLIVEC (NOVARTIS)	Capsules	Yes	Open the capsule, dilute the content in water or apple juice Attention to handling, use individual protection measures	SmPC
L01XE02	Gefitinib (250 mg)	IRESSA (ASTRAZENECA)	Coated tablets	Yes	Disperse the tablet in half glass water (it takes 20 min) Administer the suspension immediately	SmPC
L01XE03	Erlotinib hydrochloride (100, 150 mg)	TARCEVA (ROCHE)	Coated tablets	No		
L01XE04	Sunitinib maleate (12.5, 25, 50 mg)	SUTENT (PFIZER)	Capsules	Yes	Attention to handling, use individual protection measures	
L01XE05	Sorafenib tosilate (200 mg)	NEXAVAR (BAYER)	Coated tablets	No	No specific data are available	
L01XE06	Dasatinib monohydrate (50, 100, 140 mg)	BRISTOL-MYERS SQUIBB	Coated tablets	No	No specific data are available Attention to handling, use individual protection measures	
L01XE07	Lapatinib ditosylate monohydrate (250 mg)	TYVERB (NOVARTIS)	Coated tablets	No		
L01XE08	Nilotinib hydrochloride (150, 200 mg)	TASIGNA (NOVARTIS)	Capsules	Yes		
L01XE10	Everolimus (5, 10 mg)	AFINITOR (NOVARTIS)	Tablets	No		
L01XE11	Pazopanib hydrochloride (400 mg)	VOTRIENT (NOVARTIS)	Coated tablets	No		
L01XE13	Afatinib dimaleate (40 mg)	GIOTRIF (BOEHRINGER INGELHEIM)	Coated tablets	Yes	Disperse the tablet in water without breaking it, under stirring for 15 min until a fine dispersion is obtained, then administer immediately away from the EN Be careful when handling	SmPC
L01XE16	Crizotinib (250 mg)	XALKORI (PFIZER)	Capsules	No		
L01XE18	Ruxolitinib phosphate (5, 15 mg)	JAKAVI (NOVARTIS)	Tablets	No		
L01XE21	Regorafenib (40 mg)	STIVARGA (BAYER)	Coated tablets	No		
L01XX05	Hydroxycarbamide (500 mg)	ONCO CARBIDE (TEOFARMA)	Capsules	Yes	Attention to handling, use individual protection measures Antineoplastic drug	[10]
L01XX14	Tretinoin (10 mg)	VESANOID (CHEPLAPHARM ARZNEIMITTEL)	Capsules	No		
L01XX23	Mitotane (500 mg)	LYSODREN (HRA PHARMA)	Tablets	Yes	The company confirms that the tablet can be crushed and the API dissolves in water Use gloves for handling Antineoplastic drug	Manufacturer upon request
L01XX35	Anagrelide hydrochloride (0.5 mg)	XAGRID (SHIRE)	Capsules	No		
L02AB01	Megestrol acetate (160 mg)	GESTROLTEX (PHARMATEX)	Tablets	No		
L02BA01	Tamoxifen citrate (20 mg)	NOMAFEN (ITALIAN DEVICES)	Coated tablets	No		
		KESSAR (ORION)	Tablets			
L02BB01	Flutamide (250 mg)	FLUTAMIDE FDI (FIDIA FARMACEUTICI)	Tablets	Yes	Tablet disaggregates slowly, maintain it under stirring in 10 mL water for 10 min Attention to handling	[10]
L02BB03	Bicalutamide (50, 150 mg)	CASODEX (ASTRAZENECA)	Coated tablets	Yes	If possible use other antiandrogens administered as a subcutaneous implant Grind the tablet and disperse the powder in water Work in a closed system (disperse the tablet in the body of a syringe) Administer immediately after the EN Attention to handling	[10]
	Bicalutamide (150 mg)	BICALUTAMIDE HIK (HIKMA FARMACEUTICA)	Coated tablets	No	If possible use other antiandrogens administered by subcutaneous implant	[10]
L02BB03	Bicalutamide (50, 150 mg)	BICALUTAMIDE TEV (TEVA)	Coated tablets	No	If possible use other antiandrogens administered by subcutaneous implant	
L02BG03	Anastrozole (1 mg)	ANASTROZOLO ACC (ACCORD HEALTHCARE)	Coated tablets	No	Data not available	
		ARIMIDEX (ASTRAZENECA)		Yes	Disperse the tablet in 10 mL water in a closed system (disperse the tablet in the body of a syringe) Drug slightly soluble in water	
L02BG04	Letrozole (2.5 g)	LETROZOLO ACC (ACCORD HEALTHCARE)	Coated tablets	No		
		LETROZOLO SAN (SANDOZ)				
L02BG06	Exemestane (25 mg)	AROMASIN (PFIZER)	Coated tablets	No		
L02BX03	Abiraterone acetate (25 mg)	ZYTIGA (JANSSEN CILAG)	Coated tablets	No		
L04AA06	Mycophenolate sodium (180, 360 mg)	MYFORTIC (NOVARTIS)	Gastro-resistant tablets	No	Alternative: Mycofenolate mofetil (Cellept) oral solution	[8,9]
	Mycofenolate mofetil (250, 500 mg)	CELLCEPT (ROCHE)	Capsules			
	Mycofenolate mofetil (250, 500 mg)	MYFENAX (TEVA)	Coated tablets			
L04AA10	Sirolimus (0.5, 1 mg)	RAPAMUNE (PFIZER)	Coated tablets	No	Alternative: Sirolimus (Rapamune) oral solution	[8,9]
L04AA13	Leflunomide (20 mg)	ARAVA (SANOFI)	Coated tablets	Yes	Dissolve the tablet in 10 mL water and administer immediately	[10]
L04AA18	Everolimus (0.25, 0.75 mg)	CERTICAN (NOVARTIS)	Tablets	No	Alternative: orosoluble tablets to be dissolved in 10 mL water Stop EN 1 h before and 2 h after drug administration	[8,9]
L04AA27	Fingolimod hydrochloride (0.5 mg)	GILENYA (NOVARTIS)	Capsules	No		
L04AA31	Teriflunomide (14 mg)	AUBAGIO (GENZYME)	Coated tablets	No		
L04AD01	Cyclosporine (10, 25, 50, 100 mg)	SANDIMMUN NEORAL (NOVARTIS)	Capsules	No	Alternative: Cyclosporine oral solution	[8,9]
L04AD02	Tacrolimus (1 mg)	TACROLIMUS ACC (ACCORD HEALTHCARE)	Capsules	No	Alternatives: Porgraf capsules or Tacni capsules	
	Tacrolimus monohydrate (0, 5, 1, 3, 5 mg)	ADVAGRAF (ASTELLAS)				
	Tacrolimus monohydrate (0, 5, 1, 5 mg)	PROGRAF (ASTELLAS)		Yes	Open the capsule and dissolve the content in water Administer through an 8 Fr caliber tube Do not use a PVC tube Beware of handling Take on an empty stomach or at least 1 h before or 2–3 h after eating	
	Tacrolimus (0.5 mg)	TACROLIMUS MYL (MYLAN)		No	Alternative: Porgraf capsules	
	Tacrolimus monohydrate (1 mg)	ADOPORT (SANDOZ)		No	Alternative: Porgraf capsules	
L04AX01	Azathioprine (50 mg)	AZATIOPRINA (ASPEN PHARMA)	Coated tablets	Yes	Disperse the tablet in 10 mL water in the body of a syringe (closed system) Cytotoxic drug Use personal protective equipment	[10]
L04AX02	Thalidomide (50 mg)	THALIDOMIDE CELGENE (CELGENE)	Capsules	No	The capsules must not be opened Teratogen drug	
L04AX04	Lenalomide (5, 10, 15, 25 mg)	REVLIMID (CELGENE)	Capsules	No	The capsules must not be opened Teratogen drug	
L04AX05	Pirfenidone (267 mg)	ESBRIET (ROCHE)	Capsules	No		
L04AX06	Pomalidomide (4 mg)	IMNOVID (CELGENE)	Capsules	No	Not available data on drug manipulation	
M01AB01	Indomethacin (25, 50 mg)	INDOXEN (SIGMATAU)	Capsules	No	Alternative: suppositories	[8,9]
M01AB05	Diclofenac sodium (50, 150 mg)	FLOGOFENAC (A.MENARINI)	Modified release capsules	No	Alternatives: dispersible tablets, solution for injection, cutaneous formulations	[8,9]
M01AB05	Diclofenac sodium (50, 150 mg)	DICLOREUM (ALFA WASSERMANN)	Gastro-resistant tablets	No	Alternatives: dispersible tablets, solution for injection, cutaneous formulations	[8,9]
M01AB15	Ketorolac tromethamine (10 mg)	TORADOL (RECORDATI)	Coated tablets	No	Alternatives: drops, solution for injection	[8,9]
M01AC01	Piroxicam (20 mg)	FELDENE SOL (PFIZER)	Soluble tablets	Yes		[8,9]
M01AE01	Ibuprofen (600 mg)	BRUFEN (BGP PRODUCTS)	Coated tablets	No	Alternatives: effervescent granules, oral suspension to be diluted with the same water amount	[8,9]
	Ibuprofen (600 mg)	BRUFEN (BGP PRODUCTS)	Effervescent granules	Yes	Dissolve the granules in plenty of water	
	Ibuprofen (200 mg)	NUROFEN (RECKITT BENCKISER)	Coated tablets	No	Alternative: oral suspension	
	Ibuprofen arginine salt (400 mg)	SPIDIDOL (ZAMBON)	Coated tablets	No	Alternative: oral suspension	
M01AE03	Ketoprofen lysine salt (80 mg)	OKI (DOMPE’ FARMACEUTICI)	Granules for oral solution	Yes	Alternatives: solution for injection, suppositories, cutaneous gel	[8,9]
	Ketoprofen (200 mg)	KETOPROFENE EG (EG)	Capsules	No		
	Ketoprofen lysine salt (80 mg)	KETOPROFENE MYL (MYLAN)	Powder for oral solution	Yes		
	Ketoprofen (200 mg)	ORUDIS RETARD (SANOFI)	Capsules	No		
M01AH01	Celecoxib (200 mg)	CELEBREX (PFIZER)	Capsules	Yes	Open the capsule and disperse the content in 10 mL water	[10]
M01AH05	Etoricoxib (60, 90 mg)	TAUXIB (ADDENDA)	Coated tablets	Yes	The tablet immersed in 10 mL water, swells and then releases fine granules that must be administered in an 8 Fr caliber tube	[10]
	Etoricoxib (60 mg)	ARCOXIA (MSD)				
	Etoricoxib (60, 90 mg)	ALGIX (NEOPHARMED GENTILI)				
M01AX17	Nimesulide (100 mg)	AULIN (HELSINN BIREX PHARMAC.)	Granules for oral solution	Yes	Disperse the tablet in 30 mL water and then administer immediately Administer together with EN to reduce gastrointestinal side effects	
M03BX01	Baclofen (10, 25 mg)	LIORESAL (NOVARTIS)	Tablets	Yes	Grind the tablet and dissolve the powder in 10 mL water, then administer immediately	[10]
M03BX03	Pridinol mesilate (4 mg)	LYSEEN (GLAXOSMITHKLINE)	Tablets	Yes	Alternative: solution for injection	
M03BX05	Thiocolchicoside (4 mg)	MUSCORIL (SANOFI)	Capsules	No	Alternative: solution for injection	[8,9]
M04AA01	Allopurinol (100, 300 mg)	ALLOPURINOLO TEV (TEVA)	Tablets	Yes	Grind the tablet and dissolve the powder in 10 mL water, then administer immediately	[10]
M04AA03	Febuxostat (80 mg)	ADENURIC (A.MENARINI)	Coated tablets	No	Alternative: Allopurinol tablets	
M04AC01	Colchicine (1 mg)	COLCHICINA LIRCA (PHARMAFAR)	Tablets	Yes	Grind the tablet and dissolve the powder in 10 mL water, then administer immediately	[10]
M05BA04	Alendronate sodium (10 mg)	ALENDROS (ABIOGEN PHARMA)	Tablets	No	Interaction with EN Irritant for mucosal membranes If necessary grind the tablet and dissolve the powder in 50 mL water, then administer 30 min before the first daily administration of EN keeping the patient in a semi-sitting position Rinse the tube with 100 mL water after administration	
	Alendronate sodium trihydrate (70 mg)	ADRONAT (NEOPHARMED GENTILI)				
	Alendronate sodium (10 mg)	DRONAL (SIGMATAU)				
	Alendronate sodium monohydrate (70 mg)	ALENDRONATO TEV (TEVA)				
M05BA06	Ibandronate sodium monohydrate (150 mg)	BONVIVA (ROCHE)	Coated tablets	No		
		AC.IBANDR.TEV (TEVA)				
M05BB03	Alendronate sodium trihydrate Cholecalciferol (70 mg + 5600 IU)	FOSAVANCE (MSD)	Tablets	No		
M05BX03	Strontium ranelate (2 g)	PROTELOS (SERVIER)	Granules for oral suspension	Yes		
N02AA01	Morphine sulfate (10, 30 mg)	MS CONTIN (MUNDIPHARMA)	Modified release tablets	No	Alternative: drops to be diluted with 50 mL water and administered immediately	[8,9]
N02AA03	Hydromorphone hydrochloride (4, 8, 16 mg)	JURNISTA (JANSSEN CILAG)	Modified release tablets	No		[8,9]
N02AA05	Oxycodone hydrochloride (5, 10, 20 mg)	OXYCONTIN (MUNDIPHARMA)	Modified release tablets	No		[8,9]
	Oxycodone hydrochloride (40 mg)	OXICODONE SAN (SANDOZ)				
N02AA55	Oxycodone hydrochloride Paracetamol (10 + 325/20 + 325 mg)	DEPALGOS (MOLTENI & C. F.LLI ALITTI)	Coated tablets	No		[8,9]
N02AA55	Oxycodone hydrochloride Naloxone hydrochloride dihydrate (10 + 325/20 + 325 mg)	TARGIN (MUNDIPHARMA)	Modified release tablets	No	Alternative: switch to another analgesic opioid transdermal formulation	[8,9]
N02AA59	Paracetamol Codeine phosphate (500 + 30 mg)	ANGELINI	Effervescent granules	Yes	Administer only after the end of the effervescence	
			Coated tablets		The tablet can be triturated and dispersed in water	
N02AB03	Fentanyl citrate (100, 200, 300, 400, 600 µg)	ABSTRAL (PROSTRAKAN)	Sublingual tablets	No		[8,9]
	Fentanyl citrate (100, 200, 300, 400 µg)	EFFENTORA (TEVA)	Orosoluble tablets	Yes		
N02AX02	Tramadol hydrochloride (100 mg)	CONTRAMAL (GRUNENTHAL)	Modified release tablets	No	Alternatives: suppositories, orosoluble tablets, suppositories, drops to be diluted with 50 mL water	[8,9]
N02AX06	Tapentadol hydrochloride (50, 100 mg)	PALEXIA (GRUNENTHAL)	Modified release tablets	No		[8,9]
N02AX52	Tramadol hydrochloride Paracetamol (37.5 + 325 mg)	PATROL (ALFA WASSERMANN)	Coated tablets	No	Alternative: effervescent tablets to be dissolved in water and administered after the end of the effervescence	[8,9]
		KOLIBRI (ALFA WASSERMANN)				
N02BA01	Acetylsalicylic acid (100 mg)	ASPIRINETTA (BAYER)	Tablets	Yes	Immerse directly the tablet in water Administer together with EN to reduce gastrointestinal side effects	SmPC
N02BE01	Paracetamol (500 mg)	TACHIPIRINA (ANGELINI)	Tablets	Yes	Grind the tablet and disperse the powder in 10 mL water, then administer immediately Alternatives: orosoluble tablets, granules, suppositories, syrup or solution for injection	
	Paracetamol (1000 mg)		Effervescent tablets		Dissolve the tablet in 30 mL water and administer only at the end of the effervescence Alternatives: orosoluble tablets, granules, suppositories or solution for injection	
N02BE01	Paracetamol (250, 500 mg)	TACHIPIRINA FLASHTAB (ANGELINI)	Orodispersible tablets	Yes		[8,9]
	Paracetamol (250, 1000 mg)	TACHIPIRINA OROS (ANGELINI)	Orosoluble granules			
N02CC01	Sumatriptan succinate (50 mg)	SUMATRIPTAN DOC (DOC GENERICI)	Coated tablets	Yes	Alternatives: solution for injection or nasal spray	[8,9]
N03AA02	Phenobarbital (100 mg)	LUMINALE BRACCO (BRACCO)	Tablets	Yes	Grind the tablet and disperse the powder in 10 mL water, then administer immediately Administer together with EN to reduce gastrointestinal side effects Do not administer solution for injection via tube due to the presence of glycol	SmPC
N03AA02	Phenobarbital (100 mg)	GARDENALE (SANOFI)	Tablets	Yes	Grind the tablet and disperse the powder in 10 mL water, then administer immediately Administer together with EN to reduce gastrointestinal side effects Do not administer solution for injection via tube due to the presence of glycol	
N03AA03	Primidone (250 mg)	MYSOLINE (SIT LABORATORIO FARMACEUTICO)	Tablets	Yes	Grind the tablet and disperse the powder in 10 mL water, then administer immediately	
N03AB02	Phenytoin sodium (100 mg)	DINTOINA (RECORDATI)	Coated tablets	Yes	Grind the tablet and disperse the powder in 10 mL water, then administer immediately Stop EN 2 h before and 2 h after drug administration Wash the tube with 50 mL water Alternative: solution for injection	[10]
N03AB52	Phenytoin Methylphenobarbital (100 + 40 mg)	DINTOINALE (RECORDATI)	Tablets	Yes		
N03AE01	Clonazepam (2 mg)	RIVOTRIL (ROCHE)	Tablets	Yes	Alternative: Clonazepam drops to be diluted with 50 mL water and administered immediately	
N03AF01	Carbamazepine (200, 400 mg)	CARBAMAZEPINA (EG)	Tablets	Yes	Grind the tablet and disperse the powder in 10 mL sterile water, then administer immediately Stop EN 1 h before and 2 h after drug administration Requires close monitoring, it interacts with the EN Can adhere to the walls of the tube and have a reduced absorption Better to use syrup	[10]
N03AF01	Carbamazepine (400 mg)	TEGRETOL (NOVARTIS)	Modified release tablets	No	Alternatives: syrup or tablets	[8,9]
N03AF01	Carbamazepine (400 mg)	TEGRETOL (NOVARTIS)	Tablets	Yes	Grind the tablet and disperse the powder in 10 mL sterile water, then administer immediately Stop EN at least 1 h before and 2 h after the dose Requires close monitoring, it interacts with the EN Can adhere to the walls of the tube and have a reduced absorption Better to use syrup	[10]
N03AF02	Oxcarbazepine (300, 600 mg)	TOLEP (NOVARTIS)	Tablets	Yes	Grind the tablet and disperse the powder in 10 mL water, then administer immediately	
N03AG01	Valproic acid Valproate sodium (300, 500 mg)	AC VAL/S.VALP.RAT (RATIOPHARM)	Modified release tablets	No	Alternatives: oral solution or drops to be diluted with 100 mL water	[8,9]
	Valproic acid Valproate sodium (500 mg)	AC VALPROICO/SOD VALPR SAN (SANDOZ)	Modified release tablets			
	Valproate sodium (200, 500 mg)	DEPAKIN (SANOFI)	Gastro-resistant tablets			
	Valproic acid Valproate sodium (100, 250, 500, 750, 1000 mg)	DEPAKIN (SANOFI)	Modified release granules			
	Valproate magnesium (500 mg)	DEPAMAG (SIGMATAU)	Gastro-resistant tablets			
N03AG02	Valpromide (300 mg)	DEPAMIDE (SANOFI)	Gastro-resistant tablets	No		[8,9]
N03AG04	Vugabatrin (500 mg)	SABRIL (SANOFI)	Coated tablets	No	Alternative: granules	
N03AX09	Lamotrigine (25, 200 mg)	LAMOTRIGINA EG (EG)	Dispersible tablets	Yes	Dissolve the tablet in 10 mL water and administer the solution immediately	
N03AX09	Lamotrigine (25, 50, 100, 200 mg)	LAMICTAL (GLAXOSMITHKLINE)	Dispersible tablets	Yes		SmPC
N03AX11	Topiramate (25 mg)	TOPIRAMATO DOC (DOC GENERICI)	Coated tablets	Yes	The tablet is not easily dispersed in water, due to the coating, but requires gentle stirring for 5 min A fine suspension is obtained which should be administered immediately Better to use capsules	[10]
	Topiramate (25, 50, 100 mg)	TOPIRAMATO EG (EG)				
	Topiramate (25, 50, 100 mg)	TOPAMAX (JANSSEN CILAG)				
	Topiramate (25 mg)	TOPIRAMATO SAN (SANDOZ)				
N03AX11	Topiramate (25 mg)	TOPAMAX (JANSSEN CILAG)	Capsules	Yes	Open the capsule, disperse the content in 20 mL water, then administer immediately	SmPC
N03AX12	Gabapentin (400 mg)	DOC GENERICI	Capsules	Yes	Open the capsule, disperse the content in 20 mL water, then administer immediately	[10]
	Gabapentin (100, 300, 400 mg)	GABAPENTIN TEV (TEVA)				
N03AX14	Levetiracetam (500 mg)	LEVETIRACETAM DOC (DOC GENERICI)	Coated tablets	Yes	Grind the tablet and dissolve the powder in 20 mL water, then administer immediately Better to use oral solution	[10]
	Levetiracetam (500, 1000 mg)	LEVETIRACETAM SAN (SANDOZ)				
N03AX15	Zonisamide (50, 100 mg)	ZONEGRAN (EISAI)	Capsules	Yes		
N03AX16	Pregabalin (25, 75, 150 mg)	LYRICA (PFIZER)	Capsules	Yes	Open the capsules and dissolve the content in water Drug water soluble	
N03AX18	Lacosamide (50, 100 mg)	VIMPAT (UCB PHARMA)	Coated tablets	Yes	Alternative: solution for injection	
N04AA02	Biperiden hydrochloride (4 mg)	AKINETON (SIT LABORATORIO FARMACEUTICO)	Modified release tablets	No	Alternative: Biperiden hydrochloride 2 mg	[8,9]
N04AA02	Biperiden hydrochloride (2 mg)	AKINETON (SIT LABORATORIO FARMACEUTICO)	Tablets	Yes	Grind the tablet and dissolve the powder in 10 mL water, then administer immediately	
N04AA11	Bornaprine hydrochloride (4 mg)	SORMODREN (TEOFARMA)	Tablets	Yes	Grind the tablet and dissolve the powder in 10 mL water, then administer immediately Wash the tube with 50 mL water	
N04BA02	Levodopa Carbidopa (100 + 25/200 + 50 mg)	SINEMET (MSD)	Modified release tablets	No	Alternative: dispersible tablets	[8,9]
N04BA02	Levodopa Carbidopa (100 + 25/250 + 25 mg)	SINEMET (MSD)	Tablets	Yes	Grind the tablet and dissolve the powder in 10 mL water, then administer immediately Stop EN 2 h before and 2 h after drug administration Do not administer together with high-protein diets Photosensitive drug	[10]
N04BA02	Levodopa Benserazide hydrochloride (100 + 25 mg)	MADOPAR (ROCHE)	Modified release capsules	No	Alternative: dispersible tablets	[8,9]
	Levodopa Benserazide hydrochloride (100 + 25 mg)	MADOPAR (ROCHE)	Capsules			
	Levodopa Benserazide hydrochloride (100 + 25 mg)	MADOPAR (ROCHE)	Tablets			
N04BA02	Levodopa Benserazide hydrochloride (100 + 25 mg)	MADOPAR (ROCHE)	Dispersible tablets	Yes	Disperse the tablet in 25–50 mL water until a fine opalescent suspension is obtained Stop EN 1 h before drug administration	SmPC
N04BA02	Levodopa Carbidopa (200 + 50 mg)	LEVOD/CARB HEXAL (SANDOZ)	Modified release tablets	No	Alternative: dispersible tablets	[8,9]
N04BA02	Levodopa Benserazide hydrochloride (200 + 50 mg)	LEVODOPA/BENSERAZIDE TEV (TEVA)	Tablets	Yes	Grind the tablet and dissolve the powder in 10 mL water, then administer immediately Stop EN 2 h before and 2 h after drug administration Do not administer together with high-protein diets Splittable tablet	[10]
N04BA03	Levodopa Carbidopa Entecapone (all dosages)	STALEVO (NOVARTIS)	Coated tablets	No		[8,9]
N04BA05	Melevodopa hydrochloride Carbidopa hydrate (12.5 + 125/25 + 100 mg)	SIRIO (CHIESI FARMACEUTICI)	Effervescent tablets	Yes	Dissolve in 150 mL water and wait until the end of the effervescence	
N04BB01	Amantadine hydrochloride (100 mg)	MANTADAN (BOEHRINGER INGELHEIM)	Tablets	Yes	Grind the tablet and dissolve the powder in 10 mL water, then administer immediately	[10]
N04BC04	Ropirinole hydrochloride (4, 8 mg)	REQUIP (GLAXOSMITHKLINE)	Modified release tablets	No	Alternative: Requip tablets	[8,9]
N04BC04	Ropirinole hydrochloride (0.25, 0.5, 1, 2 mg)	REQUIP (GLAXOSMITHKLINE)	Coated tablets	Yes	Dissolve quickly the tablet in 10 mL water to obtain a fine dispersion to be administered immediately	[10]
N04BC04	Ropirinole hydrochloride (4 mg)	ROPINIROLO SAN (SANDOZ)	Modified release tablets	No	Alternative: Requip tablets	[8,9]
N04BC05	Pramipexole dihydrochloride monohydrate (0.26, 0.52, 1.05, 2.1, 3.15 mg)	MIRAPEXIN (BOEHRINGER INGELHEIM)	Modified release tablets	No	Alternative: Pramipexol tablets	[8,9]
N04BC05	Pramipexole dihydrochloride monohydrate (0.7 mg)	PRAMIPEXOLO EG (EG)	Tablets	Yes	Grind the tablet and dissolve the powder in 10 mL water, then administer immediately Photosensitive drug	
	Pramipexole dihydrochloride monohydrate (0.18, 0.7 mg)	PRAMIPEXOLO (TEVA)				
N04BC06	Cabergoline (2 mg)	CABASER (PFIZER)	Tablets	Yes	Grind the tablet and dissolve the powder in 10 mL water, then administer immediately	[10]
N04BD01	Selegiline hydrochloride (5, 10 mg)	JUMEX (CHIESI FARMACEUTICI)	Tablets	Yes	Grind the tablet and dissolve the powder in 10 mL water, then administer immediately Administer in the morning before starting EN	
N04BD02	Rasagiline mesylate (1 mg)	AZILECT (TEVA)	Tablets	Yes		
N04BX01	Tolcapone (100 mg)	TASMAR (MEDA PHARMA)	Coated tablets	Yes		
N04BX02	Entacapone (200 mg)	COMTAN (NOVARTIS)	Coated tablets	Yes	Grind the tablet and dissolve the powder in 20 mL water, then administer immediately	[10]
N05AA01	Chlorpromazine hydrochloride (100 mg)	PROZIN (IST.LUSOFARMACO)	Coated tablets	No	Alternatives: drops or oral solution	[8,9]
N05AA02	Levomepromazine maleate (25, 100 mg)	NOZINAN (SANOFI)	Coated tablets	No		
N05AB03	Perfenazine (2, 4, 8 mg)	TRILAFON (NEOPHARMED GENTILI)	Coated tablets	Yes		
N05AB06	Trifluoperazine dihydrochloride (1 mg)	MODALINA (IST LABORATORIO FARMACEUTICO)	Coated tablets	No		
N05AD01	Haloperidol (1, 5 mg)	HALDOL (JANSSEN CILAG)	Tablets	Yes	Better to use drops diluted in 50 mL water and administered 1 h before or 2 h after EN	
N05AF05	Zuclopenthixol dihydrochloride (10 mg)	CLOPIXOL (LUNDBECK)	Coated tablets	Yes	Immerse the tablet in water, after 10 min a fine suspension is obtained, then administer immediately Alternative: solution for injection	
N05AH02	Clozapine (25 mg)	CLOZAPINA CHS (CHIESI FARMACEUTICI)	Coated tablets	Yes	Grind the tablet and dissolve the powder in 10 mL water, then administer immediately	
	Clozapine (25, 100 mg)	LEPONEX (NOVARTIS)	Tablets			
	Clozapine (100 mg)	CLOZAPINA (ORION)	Tablets			
N05AH03	Olanzapine (5, 10 mg)	ZYPREXA VELOTAB (ELI LILLY)	Orodispersible tablets	Yes	Dissolve in 30 mL water, apple juice, orange juice, or milk Irritant drug Use personal protective equipment	SmPC
N05AH03	Olanzapine (2.5 mg)	ZYPREXA (ELI LILLY)	Coated tablets	Yes	Better to use orodispersible tablets Irritant to mucous membranes and eyes	
	Olanzapine (5, 10 mg)	OLANZAPINA SUN (SUN PHARMACEUTICALS)	Orodispersible tablets	Yes	Dissolve the tablet in water and administer immediately	
	Olanzapine (2.5, 5, 10 mg)	OLANZAPINA TEV (TEVA)	Coated tablets	No	Better to use orodispersible tablets	
N05AH03	Olanzapine (5, 10 mg)	OLANZAPINA TEV (TEVA)	Orodispersible tablets	Yes	Dissolve the tablet in water and administer immediately	SmPC
N05AH04	Quetiapine fumarate (50, 150, 200, 300, 400 mg)	SEROQUEL (ASTRAZENECA)	Modified release tablets	No	Alternative: Seroquel coated tablets	[8,9]
N05AH04	Quetiapine fumarate (100, 200, 300 mg)	SEROQUEL (ASTRAZENECA)	Coated tablets	Yes	Grind the tablet and dissolve the powder in 25 mL water, then administer immediately	
	Quetiapine fumarate (50, 200, 300, 400 mg)	QUETIAPINA TEV (TEVA)		No		
N05AH04	Quetiapine fumarate (50, 200, 300, 400 mg)	QUETIAPINA TEV (TEVA)	Modified release tablets	No		[8,9]
N05AH05	Asenapine maleate (5, 10 mg)	SYCREST (LUNDBECK)	Sublingual tablets	No		[8,9]
N05AH06	Clotiapine (40 mg)	ENTUMIN (LAB.JUVISE’ PHARMACEUTICALS)	Tablets	Yes	Grind the tablet and dissolve the powder in 10 mL water, then administer immediately Alternative: Entumin drops to be diluted with 50 mL water	
N05AL03	Tiapride hydrochloride (100 mg)	SEREPRILE (SANOFI)	Tablets	Yes	Grind the tablet and dissolve the powder in 10 mL water, then administer immediately Administer at a fixed time	
N05AL05	Amilsupride (50 mg)	AMISULPRIDE EG (EG)	Tablets	Yes	Alternative: oral solution	[10]
	Amilsupride (50, 200, 400 mg)	AMISULPRIDE MYL (MYLAN)	Tablets			
	Amilsupride (400 mg)	AMISULPRIDE SAN (SANDOZ)	Coated tablets			
	Amilsupride (400 mg)	SOLIAN (SANOFI)	Coated tablets			
N05AL07	Levosulpride (50, 100 mg)	LEVOPRAID (TEOFARMA)	Tablets	Yes	Grind the tablet and dissolve the powder in 10 mL water, then administer immediately	[10]
N05AN01	Lithium carbonate (300 mg)	LITIO CARBONATO (NOVA ARGENTIA)	Tablets	Yes	Grind the tablet and dissolve the powder in 10 mL water, then administer immediately Attention drug with a narrow therapeutic index	
		CARBOLITHIUM (TEVA)	Capsules			
N05AX08	Risperidone (1, 2 mg)	RISPERIDONE SAN (SANDOZ)	Coated tablets	No	Alternative: liquid formulation	[8,9]
	Risperidone (2, 3, 4 mg)	RISPERIDONE TEV (TEVA)				
N05AX12	Aripripazole (5, 10, 15 mg)	ABILIFY (OTSUKA)	Tablets	Yes	Alternatives: orodispersible tablets or oral solution	[8,9]
	Aripripazole (10, 15 mg)	ABILIFY (OTSUKA)	Orodispersible tablets			[8,9]
N05AX13	Paliperidone (3, 6, 9 mg)	INVEGA (JANSSEN CILAG)	Modified release tablets	No	Alternative: Xeplion modified release suspension for injection	[8,9]
N05BA	Delorazepam (0.5, 1, 2 mg)	DELORAZEPAM (AUROBINDO)	Tablets	No	Alternative: drops	[8,9]
	Delorazepam (0.5, 1, 2 mg)	EN (BGP PRODUCTS)				
	Delorazepam (2 mg)	DELORAZEPAM RAT (RATIOPHARM)				
	Delorazepam (0.5, 1 mg)	DELORAZEPAM WPI (ZENTIVA)				
N05BA01	Diazepam (2 mg)	VALIUM (ROCHE)	Capsules	Yes	Open the capsule, disperse the content in 10 mL water, then administer immediately	[10]
	Diazepam (5 mg)	VATRAN (VALEAS)	Tablets		Disperse the tablet in 10 mL water, then administer immediately	
N05BA04	Oxazepam (15 mg)	SERPAX (MEDA PHARMA)	Tablets	Yes	Grind the tablet and disperse it in 10 mL water, then administer immediately Divisible tablet Better to use Diazepam or Lorazepam	[10]
N05BA06	Lorazepam (1, 2.5 mg)	TAVOR (PFIZER)	Tablets	Yes	Disperse the tablet in 10 mL water, then administer immediately Prefer liquid or the orosoluble formulation	[10]
N05BA08	Bromazepam (1.5 mg)	COMPENDIUM (POLIFARMA)	Capsules	No	Alternative: drops	[8,9]
	Bromazepam (3 mg)	BROMAZEPAM RAT (RATIOPHARM)	Tablets			
N05BA09	Clobazam (10 mg)	FRISIUM (SANOFI)	Capsules	Yes	Open the capsule, disperse the content in 10 mL water, then administer immediately	[10]
N05BA11	Prazepam (10 mg)	PRAZENE (PFIZER)	Tablets	No	Alternative: drops that must be diluted	[8,9]
N05BA12	Alprazolam (0.25, 0.5 mg)	XANAX (PFIZER)	Tablets	Yes	Disperse the tablet in 10 mL water, then administer immediately Better to use the drops	[10]
N05BB01	Hydroxyzine hydrochloride (25 mg)	ATARAX (UCB PHARMA)	Coated tablets	Yes	Prefer Atarax syrup	[10]
N05CD01	Flurazepam monohydrochloride (15, 30 mg)	VALDORM (VALEAS)	Capsules	Yes	Open the capsule, disperse the content in 20 mL water, then administer immediately Administer before the last EN	[10]
N05CD02	Nitrazepam (5 mg)	MOGADON (MEDA PHARMA)	Tablets	Yes	Dissolve the tablet in a water glass and then administer immediately	SmPC
N05CD05	Triazolam (250 µg)	HALCION (PFIZER)	Tablets	Yes		
N05CD09	Brotizolam (0.25 mg)	LENDORMIN (BOEHRINGER INGELHEIM)	Tablets	Yes	Divisible tablets	
N05CF02	Zolpidem tartrate (10 mg)	ZOLPIDEM RAT (RATIOPHARM)	Coated tablets	Yes	Grind the tablet and disperse it in 10 mL water, then administer immediately Better to administer at bedtime to have a hypnotic effect Divisible tablets	
N06AA04	Clomipramine hydrochloride (75 mg)	ANAFRANIL (SIGMATAU)	Modified release tablets	No		[8,9]
N06AA04	Clomipramine hydrochloride (10, 25 mg)	ANAFRANIL (SIGMATAU)	Coated tablets	Yes	Grind the tablet and disperse it in 10 mL water, then administer immediately	[10]
N06AA09	Amitriptyline hydrochloride (10, 25 mg)	LAROXYL (TEOFARMA)	Coated tablets	No	Alternative: drops that must be diluted with 50 mL water	[8,9]
N06AA10	Nortriptyline hydrochloride (10, 25 mg)	NORITREN (LUNDBECK)	Coated tablets	Yes		
N06AB03	Fluoxetine hydrochloride (20 mg)	FLUOXETINA EG (EG)	Capsules	Yes	Open the capsule, disperse the content in 20 mL water, then administer immediately Prefer liquid formulation or orosoluble tablets The solution should not be administered in the fast due to the acidic pH	[10]
N06AB03	Fluoxetine hydrochloride (20 mg)	FLUOXETINA FDI (FIDIA FARMACEUTICI)	Capsules	Yes	Preferably use the liquid form or the dispersible tablets	[8,9]
N06AB03	Fluoxetine hydrochloride (20 mg)	FLUOXETINA RAT (RATIOPHARM)	Soluble tablets	Yes	Dissolve the tablet in 50 mL water and administer immediately	SmPC
N06AB03	Fluoxetine hydrochloride (20 mg)	XEREDIEN (VALEAS)	Dispersible tablets	Yes	Dissolve the tablet in water and administer immediately	[8,9]
N06AB04	Citalopram hydrobromide (20 mg)	CITALOPRAM RAT (RATIOPHARM)	Coated tablets	Yes	Grind the tablet and disperse it in 10 mL water, then administer immediately Preferably use drops Incompatible with PVC tubes	[10]
		CITALOPRAM HEX (SANDOZ)				[10]
N06AB05	Paroxetine hydrochloride (20 mg)	SEREUPIN (GLAXOSMITHKLINE)	Coated tablets	Yes	Grind the tablet and disperse the powder in 10 mL water, then administer immediately Alternatives: drops or oral suspension	[10]
N06AB06	Sertraline hydrochloride (50 mg)	SERTRALINA MYL (MYLAN)	Coated tablets	Yes	Grind the tablet and disperse the powder in 10 mL water, then administer immediately Rinse the tube well after administration Preferably switch to another selective serotonin reuptake inhibitor	
		ZOLOFT (PFIZER)				
N06AB08	Fluvoxamine maleate (50 mg)	FEVARIN (BGP PRODUCTS)	Coated tablets	No	Preferably switch to another selective serotonin reuptake inhibitor in liquid formulation	[8,9]
		MAVERAL (BGP PRODUCTS)				
N06AB10	Escitalopram oxalate (10, 20 mg)	CIPRALEX (LUNDBECK)	Coated tablets	Yes	Alternative: Cipralex drops Divisible tablets	[10]
	Escitalopram oxalate (20 mg)	ENTACT (LUNDBECK)				
N06AX03	Mianserin hydrochloride (30 mg)	LANTANON (MSD)	Coated tablets	Yes	Grind the tablet and disperse the powder in 10 mL water, then administer immediately	
N06AX05	Tradozone hydrochloride (75, 150 mg)	TRITTICO (ANGELINI)	Modified release tablets	No	Alternatives: coated tablets or drops	[8,9]
N06AX05	Tradozone hydrochloride (75, 150 mg)	TRITTICO (ANGELINI)	Coated tablets	Yes	Grind the tablet and disperse the powder in 10 mL water, then administer immediately Divisible tablets	
N06AX11	Mirtazapine (30 mg)	MIRTAZAPINA SAN (SANDOZ)	Coated tablets	Yes		
N06AX12	Bupropion hydrochloride (150 mg)	WELLBUTRIN (GLAXOSMITHKLINE)	Modified release tablets	No		[8,9]
N06AX16	Venlafaxine hydrochloride (75, 150 mg)	ZARELIS (ITALFARMACO)	Modified release tablets	No	Alternative: oral solution	[8,9]
	Venlafaxine hydrochloride (37.5, 75, 150 mg)	EFEXOR (PFIZER)	Modified release capsules			
	Venlafaxine hydrochloride (37.5 mg)	VENLAFAX TEV (TEVA)	Modified release capsules			
N06AX21	Duloxetine hydrochloride (30, 60 mg)	CYMBALTA (ELI LILLY)	Modified release capsules	No	Risk of probe occlusion	[8,9]
	Duloxetine hydrochloride (60 mg)	XERISTAR (QUINTILES)	Gastro-resistant capsules	No	Do not grind the granules	[8,9]
N06BA04	Methylphenidate hydrochloride (10 mg)	RITALIN (NOVARTIS)	Tablets	Yes		
N06BA04	Methylphenidate hydrochloride (10, 20 mg)	EQUASYM (SHIRE)	Modified release capsules	No		[8,9]
N06BA07	Modafinil (100 mg)	PROVIGIL (TEVA)	Tablets	Yes	Grind the tablet and disperse the powder in water, then administer immediately	[10]
N06BA09	Atomoxetine hydrochloride (40, 60 mg)	STRATTERA (ELI LILLY)	Capsules	No		
N06BX12	Acetyl-L-carnitine (500 mg)	NICETILE (SIGMATAU)	Gastro-resistant tablets	No		[8,9]
N06BX12	Acetyl-L-carnitine (500 mg)	NICETILE (SIGMATAU)	Powder for oral solution	Yes		
N06CA01	Amitriptyline Perphenazine (10 + 4 mg)	MUTABON MITE (NEOPHARMED GENTILI)	Coated tablets	No		[8,9]
N06DA02	Donepezil hydrochloride (10 mg)	ARICEPT 10 (PFIZER)	Coated tablets	Yes	Grind the tablet and disperse the powder in 20 mL water, then administer within 15 min Preferably use Donezepil orodispersible tablets	[10]
	Donepezil hydrochloride (5, 10 mg)	DONEPEZIL ACV (ACTAVIS)				
		MEMAC (BRACCO)				
		DONEPEZIL EG (EG)				
N06DA02	Donepezil hydrochloride monohydrate (5 mg)	DONEPEZIL MYL (MYLAN)	Orodispersible tablets	Yes	Dissolve the tablet in 10 mL water and administer immediately	[8,9]
	Donepezil hydrochloride (5, 10 mg)	DONEPEZIL TEV (TEVA)				
N06DA03	Rivastigmine hydrogen tartrate (1.5, 3, 4.5 mg)	EXELON (NOVARTIS)	Capsules	No	Alternatives: transdermal patches or oral solution	[8,9]
	Rivastigmine hydrogen tartrate (1.5, 3, 4.5, 6 mg)	RIVASTIGMINA SAN (SANDOZ)				
N06DA04	Galantamine hydrobromide (8, 16, 24 mg)	REMINYL (JANSSEN CILAG)	Modified release capsules	No	Alternative: oral solution	[8,9]
N06DA04	Galantamine hydrobromide (8, 16, 24 mg)	REMINYL (JANSSEN CILAG)	Coated tablets	Yes	Grind the tablet and disperse the powder in 10 mL water, then administer Preferably use Galantamine oral solution to be diluted before administration	[10]
N06DX01	Memantine hydrochloride (10 mg)	EBIXA (LUNDBECK)	Coated tablets	No	Alternative: Ebixa oral solution	[8,9]
	Memantine hydrochloride (10, 20 mg)	MEMANTINA MYL (MYLAN)				
N07AA02	Pyridostigmine hydrobromide (180 mg)	MESTINON (MEDA PHARMA)	Modified release tablets	No	Alternative: Mestinton 60 mg	[8,9]
N07AA02	Pyridostigmine hydrobromide (60 mg)	MESTINON (MEDA PHARMA)	Tablets	Yes	Divisible tablets	
N07AX01	Pilocarpine hydrochloride (5 mg)	SALAGEN (MERUS LABS LUXCO II SARL)	Coated tablets	No		
N07BB01	Disulfiram (400 mg)	ANTABUSE DISPERGETTES (AUROBINDO)	Effervescent tablets	Yes	Dissolve the tablet in 10 mL water and administer at the end of the effervescence	
N07BB03	Acamprosate calcium (333 mg)	CAMPRAL (BRUNO FARMACEUTICI)	Coated tablets	No		
N07BC01	Buprenorphine hydrochloride (2, 8 mg)	BUPRENORFINA MOL (MOLTENI & C. F.LLI ALITTI)	Sublingual tablets	No	Administer sublingually only if the patient is conscious	[8,9]
N07CA01	Betahistine dihydrochloride (8 mg)	MICROSER (GRUNENTHAL)	Tablets	Yes	Grind the tablet and disperse the powder in 20 mL water, then administer Preferably use drops	[10]
N07CA02	Cinnarizine (75 mg)	STUGERON FTE (JANSSEN CILAG)	Capsules	No	Alternative: drops	[8,9]
		STUGERON FTE (JANSSEN CILAG)	Tablets			
N07XX02	Riluzole (50 mg)	RILUZOLO SAN (SANDOZ)	Coated tablets	Yes	Grind the tablet and disperse the powder in water, then administer Preferably change therapy	[10]
N07XX06	Tetrabenazine (25 mg)	XENAZINA (CHIESI FARMACEUTICI)	Tablets	Yes	Divisible and crushable tablet	
N07XX09	Dimethyl fumarate (120, 240 mg)	TECFIDERA (BIOGEN ITALIA)	Gastro-resistant capsules	No	Do not grind the granules	[8,9]
P01AB01	Metronidazole (250 mg)	VAGILEN (ALFA WASSERMANN)	Capsules	Yes	Open the capsule, disperse the content in 20 mL water, then administer immediately 1 h before EN Alternatives: solution for injection or vaginal ovules	
		FLAGYL (ZAMBON)	Tablets	No	Alternatives: solution for injection or vaginal ovules	
P01BA01	Chloroquine diphosphate (250 mg)	CLOROCHINA (BAYER)	Coated tablets	Yes	Grind the tablet and dissolve the powder in 10 mL water, then administer immediately preferably after EN	[10]
P01BA02	Hydroxychloroquine sulfate (200 mg)	PLAQUENIL (SANOFI)	Coated tablets	Yes		[10]
P01BF05	Piperaquine tetraphosphate Dihydroartemisinin (320 + 40 mg)	EURARTESIM (SIGMATAU)	Coated tablets	Yes		
P02CA03	Albendazole (400 mg)	ZENTEL (GLAXOSMITHKLINE)	Tablets	Yes	Grind the tablet and dissolve the powder in 10 mL water, then administer immediately	SmPC
P02DA01	Niclosamide (500 mg)	YOMESAN (BAYER)	Chewable tablets	Yes	Dissolve the tablet in water and administer	SmPC
R03DA04	Theophylline anhydrous (200, 300 mg)	THEO-DUR (RECORDATI)	Modified release tablets	No	Alternative: liquid formulation to be diluted with 20 mL water and administered 1 h after EN Monitor blood levels and adjust dosage	[8,9]
		RESPICUR (TAKEDA)	Modified release capsules			
R03DA08	Bamifylline hydrochloride (600 mg)	BAMIFIX (CHIESI FARMACEUTICI)	Coated tablets	No	Alternative: suppositories but only for pediatrics	
R03DA11	Doxofylline (400 mg)	ANSIMAR (ABC FARMACEUTICI)	Tablets	Yes	Alternatives: Ansimar syrup or solution for injection	
R03DC03	Montelukast sodium (5 mg)	SINGULAIR (MSD)	Chewable tablets	Yes		
	Montelukast sodium (10 mg)	SINGULAIR (MSD)	Coated tablets		Disperse the tablet in 10 mL water under gentle stirring, then administer immediately	
	Montelukast sodium (5 mg)	MONTELUKAST SAN (SANDOZ)	Chewable tablets			
	Montelukast sodium (5 mg)	MONTELUKAST TEV (TEVA)	Coated tablets		Grind the tablet and disperse the powder in water, then administer away from EN Preferably use oral granules	
R05CB01	Acetylcysteine (600 mg)	ACETILCISTEINA RAT (RATIOPHARM)	Effervescent tablets	Yes	Administer at the end of the effervescence	
		FLUIMUCIL (ZAMBON)	Granules for oral solution		Mix with water then administer	
		FLUIMUCIL (ZAMBON)	Effervescent tablets		Administer at the end of the effervescence	
R06AE06	Oxatomide (30 mg)	TINSET (GRUNENTHAL)	Tablets	No	Alternative: drops	[8,9]
R06AE07	Cetirizine hydrochloride (10 mg)	CERCHIO (MEDIOLANUM FARMACEUTICI)	Tablets	Yes	Alternative: drops	
		CETIRIZINA SAN (SANDOZ)	Coated tablets			
R06AX25	Mizolastine (10 mg)	ZOLISTAM (SANOFI)	Modified release tablets	No		[8,9]
R06AX26	Fexofenadine hydrochloride (180 mg)	TELFAST (SANOFI)	Coated tablets	No	Alternatives: Cetirizine or Loratadine	
R06AX27	Desloratadine (2.5 mg)	AERIUS (MSD)	Orodispersible tablets	Yes		
S01EC01	Acetazolamide (250 mg)	DIAMOX (TEOFARMA)	Tablets	Yes		
V03AC03	Deferasirox (125, 500 mg)	EXJADE (NOVARTIS)	Dispersible tablets	Yes	Disperse in 200 mL water until a suspension is obtained Take on an empty stomach before EN	SmPC
V03AE02	Sevelamer hydrochloride (800 mg)	RENAGEL (GENZYME)	Coated tablets	No	Alternative: Renvela powder for oral suspension	[8,9]
		RENVELA (GENZYME)				
V03AE02	Sevelamer carbonate (2.4 g mg)	RENVELA (GENZYME	Powder for oral suspension	Yes	Disperse the powder in 60 mL water and administer immediately Rinse well the tube	
V03AE03	Lanthanum(III) carbonate hydrate (1 g)	FOZNOL (SHIRE)	Oral powder	No		
	Lanthanum(III) carbonate hydrate (1 g)	FOZNOL (SHIRE)	Chewable tablets	Yes		
V03AE04	Calcium acetate Magnesium carbonate (435 + 235 mg)	OSVAREN (VIFOR FRESENIUS)	Coated tablets	No		
V03AF	Calcium mefolinate (15 mg)	PREFOLIC (ZAMBON)	Gastro-resistant tablets	No		[8,9]
V03AF04	Calcium levofolinate (7.5 mg)	LEDERFOLIN (PFIZER)	Tablets	Yes	Grind the tablet and dissolve the powder in 10 mL water, then administer Alternative: powder for reconstitution for injection	[10]
V03AF04	Calcium levofolinate (2.5 mg)	LEDERFOLIN (PFIZER)	Granules for oral solution	Yes		

**Table 4 jpm-12-01307-t004:** List of magistral preparations routinely compounded in the hospital pharmacy.

Magistral Preparation	Ingredients	Storage/Expiration	Method of Administration
Captopril 1 mg/mL Syrup	Captopril powder Pharm. Eur. Simple syrup	5 °C in a well-closed amber glass jar for 10 days	Shake, withdraw the required portion of the liquid and dilute with water *
Carbamazepine 40 mg/mL Suspension	Carbamazepine powder Pharm. Eur. Simple syrup		
Isoniazid 50 mg/mL Solution	Isoniazid powder Pharm. Eur. Distilled water		
Midazolam 2.5 mg/mL Syrup	Midazolam powder Pharm. Eur. Simple syrup		
Omeprazole 2 mg/mL Solution	Omeprazole powder Pharm. Eur. Sodium bicarbonate 8.4%		
Ranitidine 15 mg/mL Solution	Omeprazole powder Pharm. Eur. Sorbitol Purified water		
Spironolactone 10 mg/mL Suspension	Spironolactone powder Pharm. Eur. Simple syrup		
Verapamil 50 mg/mL Suspension	Verapamil powder Pharm. Eur. Distilled water Simple syrup Glycerol		

* 60 mL.

**Table 5 jpm-12-01307-t005:** General recommendations for medications given by tube.

Pharmaceutical Form	Method of Administration
Liquid forms (drops, syrups, suspensions)	High osmolar or very viscous liquids must be suitably diluted before being administered If the osmolality is not known, it is advisable to dilute the drug with at least 30 mL of water to make it compatible with gastric administration and to prevent diarrheal phenomena from osmotic effect Shake and mix the solution or suspension well Administer directly by tube
Tablets Chewable tablets	The tablets must be crushed in the crusher until they are reduced to a fine and homogeneous powder to promote better absorption and avoid obstruction of the tube Transfer the powder to a plastic cup
Effervescent tablets	Put the tablet directly in a plastic cup
Dispersible/soluble tablets	Put the tablet directly in a plastic cup
Capsules	Open the capsule and insert the powder in a plastic cup, verifying the content first
Soft capsules	They cannot be administered
Modified-release tablets/capsules	
Gastroresistant tablets, enteric-coated tablets	
Sublingual tablets	

## Data Availability

Data are contained within the article.

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
