# Peer review of "The Role of the Pharmacist in Selecting the Best Choice of Medication Formulation in Dysphagic Patients"

_jpm, 2022, doi:10.3390/jpm12081307_

Round 1

Reviewer 1 Report

In this submission, the author has discussed the role of the pharmacist in selecting the medication formulation for dysphagic patients. The manuscript has been organized in a proper way with a mounting body of information. I appreciate the author’s effort to summarize all the information in table 3. This submission might be more suitable as a review or case study. The manuscript can be accepted after minor corrections.

 Here are the comments which need to be addressed.

1.     The drug stability before the administration is one of the critical parameters which endow its therapeutic effect. The author mentioned that they get back from one company out of 124. However, it would be good to have a brief discussion mentioning the possible drug instabilities.   

2.     Two right-side columns in Table 4 are exactly presenting the same information for all preparation. It is suggested to remove this information and mention this information in the table caption. 

Author Response

Answers to Reviewer 1

In this submission, the author has discussed the role of the pharmacist in selecting the medication formulation for dysphagic patients. The manuscript has been organized in a proper way with a mounting body of information. I appreciate the author’s effort to summarize all the information in table 3. This submission might be more suitable as a review or case study. The manuscript can be accepted after minor corrections.

We thank the Reviewer for his/her appreciation and for understanding our great work in collecting and summarizing the data in Table 3.

The drug stability before the administration is one of the critical parameters which endow its therapeutic effect. The author mentioned that they get back from one company out of 124. However, it would be good to have a brief discussion mentioning the possible drug instabilities.   

We agree with the Reviewer so a brief discussion has been included in the Section 3.1.

Two right-side columns in Table 4 are exactly presenting the same information for all preparation. It is suggested to remove this information and mention this information in the table caption. 

We agree with the Reviewer and the changes were made accordingly.

Reviewer 2 Report

Comments to the Authors

The authors have the role of pharmacist to select a better formulation of medicines so it can be administered and swallowed easily for specific category of population. Authors have listed good number of ways. The article is interesting and valuable in the healthcare sector. It can be important for other health care professionals like doctors, nurses as well. However, I have some minor points to comment on.

·         Out of 124 pharmaceutical companies, only one company gave further information. What could be the reason of decline by the pharmaceutical companies or there was lack follow-up?

·         How different routes of administration will affect the bioavailability and bioequivalence?

·         References can be provided in the Introduction section.

·         What was the age group of patients?

·         Was randomization done?

·         Grammar can be improved in the manuscript.

·         Figure 1 is not completely legible. It can be improved.

·         In Table 1, what was the criteria for reliability?

·         Is there any possibility of intraocular and intraperitoneal route?

Author Response

Answers to Reviewer 2

The authors have the role of pharmacist to select a better formulation of medicines so it can be administered and swallowed easily for specific category of population. Authors have listed good number of ways. The article is interesting and valuable in the healthcare sector. It can be important for other health care professionals like doctors, nurses as well.

We thank the Reviewer for his/her appreciation and for understanding our goal of sharing evidence-based data and our great work in gathering all information.

Out of 124 pharmaceutical companies, only one company gave further information. What could be the reason of decline by the pharmaceutical companies or there was lack follow-up?

A brief discussion in order to explain the reason of the decline by the pharmaceutical companies has been include in the Section 3.1. The pharmaceutical companies, in fact, avoid taking any kind of responsibility in giving us some advice as they have not carried out any evaluation of the stability of the drug after handling.

How different routes of administration will affect the bioavailability and bioequivalence?

As we have extensively explained in the Results, pharmaceutical incompatibilities, pharmacological incompatibilities or pharmacokinetic interactions can alter the processes of release, absorption, distribution, metabolism and excretion of drugs. Several examples have been provided in Section 3.1, so we kindly ask the Reviewer to reread this part of the manuscript.

References can be provided in the Introduction section.

We agree with the reviewer and therefore the references in the Introduction section have been reorganized.

What was the age group of patients?

The patients were of a wide range of age, both pediatrics and adults. We have specified it where the number of patients involved was entered.

Was randomization done?

The aim of the study was not to assess the effect of the manipulated formulations on patients but to provide instructions on how to handling medicinal product to be administered by tube, it was not a clinical study therefore randomization was not applicable.

Grammar can be improved in the manuscript

We thank the Reviewer, indeed after carefully rereading the manuscript we find a lot of grammar and editing errors.

Figure 1 is not completely legible. It can be improved.

The Figure quality is in according to the Journal requirement and the text is clear only the number of protocol, telephone number, e-mail etc. appear less evident but they are not relevant to our purpose.  

In Table 1, what was the criteria for reliability?

The criteria of reliability has been established by us on the basis of the importance of the Authority which provides the information. At the higher level we put the SmPC as its content is supported by clinical experiments and it has been approved by EMA before marketing authorization, followed by European or National guidelines, then published work in the field, and finally pharmacist know-how.

Is there any possibility of intraocular and intraperitoneal route?

We thank the Reviewer for his/her advice, but unfortunately there isn’t. We point out that the medical prescriptions  should include only the medicines embedded in the hospital formulary.